# Long-term culture of human pancreatic slices as a model to study real-time islet regeneration

Mirza Muhammad Fahd Qadir [1,2], Silvia Álvarez-Cubela[1], Jonathan Weitz[3,13], Julia K. Panzer[3,13], Dagmar Klein[1], Yaisa Moreno-Hernández[1,4], Sirlene Cechin[1], Alejandro Tamayo[3], Joana Almaça[3], Helmut Hiller[5], Maria Beery [5], Irina Kusmartseva[5], Mark Atkinson[5,6], Stephan Speier [7,8,9], Camillo Ricordi[1,10], Alberto Pugliese[3,11], Alejandro Caicedo[3], Christopher A. Fraker[1,12], Ricardo Luis Pastori[1,3 ✉] & Juan Domínguez-Bendala [1,2,10 ✉]

The culture of live pancreatic tissue slices is a powerful tool for the interrogation of physiology and pathology in an in vitro setting that retains near-intact cytoarchitecture. However, current culture conditions for human pancreatic slices (HPSs) have only been tested for short-term applications, which are not permissive for the long-term, longitudinal study of pancreatic endocrine regeneration. Using a culture system designed to mimic the physiological oxygenation of the pancreas, we demonstrate high viability and preserved endocrine and exocrine function in HPS for at least 10 days after sectioning. This extended lifespan allowed us to dynamically lineage trace and quantify the formation of insulin-producing cells in HPS from both non-diabetic and type 2 diabetic donors. This technology is expected to be of great impact for the conduct of real-time regeneration/developmental studies in the human pancreas.

[1] Diabetes Research Institute, University of Miami Miller School of Medicine, Miami, FL 33136, USA. [2] Department of Cell Biology and Anatomy, University of Miami Miller School of Medicine, Miami, FL 33136, USA. [3] Department of Medicine, Division of Metabolism, Endocrinology and Diabetes, University of Miami Miller School of Medicine, Miami, FL 33136, USA. [4] Universidad Francisco de Vitoria, Madrid, Spain. [5] nPOD Laboratory, Department of Pathology, Immunology and Laboratory Medicine, University of Florida, Gainesville, FL 32611, USA. [6] Department of Pathology, Immunology and Laboratory Medicine, College of Medicine, University of Florida, Gainesville, FL 32611, USA. [7] Paul Langerhans Institute Dresden (PLID) of the Helmholtz Zentrum München at the University Clinic Carl Gustav Carus of Technische Universität Dresden, Helmholtz Zentrum München, Neuherberg, Germany. [8] Faculty of Medicine, Institute of Physiology, Technische Universität Dresden, Dresden, Germany. [9] German Center for Diabetes Research (DZD), München, Neuherberg, Germany. [10] Department of Surgery, University of Miami Miller School of Medicine, Miami, FL 33136, USA. [11] Department of Microbiology & Immunology, University of Miami Miller School of Medicine, Miami, FL 33136, USA. [12] Department of Biomedical Engineering, University of Miami Miller School of Medicine, Miami, FL 33136, USA. [13] These authors contributed equally: Jonathan Weitz, Julia K. Panzer. ✉email: rpastori@med.miami.edu; jdominguez2@med.miami.edu

The identification of methods to stabilize, section and culture slices of live pancreatic tissue[1,2] represented a qualitative leap in our ability to study the biology and function of the pancreas. This in vitro setting preserves much of the cytoarchitecture of the organ, allowing for the dynamic study of islet physiology and the interactions between endocrine, exocrine, neural, vascular and immune cells in their anatomical context[3–7]. Since this method is less damaging to the tissue than those used to isolate islets or exocrine tissue, the studied phenomena are more likely to approximate the native biology of the organ. As originally reported, the method entails the retrograde injection of low melting point (LMP) agarose through the major pancreatic duct of mice[1,4]. LMP hardens and stabilizes the pancreas, which can be subsequently extracted, further embedded in agarose and sliced with a vibratome. Slicing techniques have recently been adapted to human pancreas[8,9]. These reports have attracted widespread interest, especially owing to simultaneous initiatives such as that of the Network of Pancreatic Organ Donors with Diabetes (nPOD) program, to obtain and distribute live pancreatic tissue from type 1 diabetes (T1D) patients. The use of slicing techniques on those precious samples has the potential to unveil a wealth of information on the dynamic processes of autoimmunity and regeneration in T1D. However, current settings (which entail the culture of slices atop transwell inserts) have not been tested for long-term culture, a sine qua non requirement for the study of regeneration. Based on our previous experience with human isolated islets[10] and embryonic-stem cell (hESc)-derived β-like cells[11,12], we anticipated that improper oxygenation might induce hypoxia and thus compromise HPS viability and function over several days of culture. This was confirmed experimentally through metabolic and viability assays. Therefore, we sought to identify improved ways to culture HPSs long-term. Using a perfluorocarbon (PFC)-based system designed to enhance oxygen transfer to three-dimensional tissues[10,11,13], as well as a refined medium composition, herein we present evidence that HPSs from both healthy and T1D/T2D donors survive and exhibit normal function beyond 10 days of culture. This extended lifespan was sufficient to detect, longitudinally track and quantify β-cell regeneration in slices from transgenic mice[14] as well as HPSs from both healthy and T2D human donors cotransduced with adenoviral vectors carrying a reporter and an insulin lineage tracer. The latter approach opens the door to the study of human pancreatic remodeling and regeneration in an in vitro setting that largely preserves the compartment distribution, cell-to-cell interactions and niches of the native organ.

## Results

### Mathematical modeling of oxygen diffusion in HPSs.
To predict oxygenation patterns across 120 μm-thick HPSs, we performed 2D diffusion reaction modeling of oxygen distribution using COMSOL v5.3 (COMSOL Inc, Burlington, MA) as described[10,11]. A parametric sweep of oxygen consumption rate (OCR) ($1.0–5.0 \times 10^{-2}$ mol/m³ s⁻¹) unique to pancreatic tissue[15] was performed. An estimated oxygen consumption rate (OCR) of 0.03 mol/m³ s⁻¹ (average for islets and acinar tissue[15]) was used (Fig. 1). Two culture settings were modeled: standard transwell[1] and PFC-based dishes[10]. While an air-liquid interface above the tissue is acceptable for short-term analyses, the medium evaporates quickly. Studies requiring survival for longer than a few hours require the slices to sit atop a liquid-permeable membrane and be surrounded by medium both from the top and the bottom (Fig. 1a, left). Unlike in conventional transwells, in the PFC setting, slices rest atop a liquid-impermeable PFC-based membrane[10,11,13] that provides direct contact with oxygen from underneath (Fig. 1a, right), whereas the tissue is completely

covered by medium. Figure 1b shows the predicted oxygenation patterns of transwell- and PFC-cultured slices throughout 24 h when incubated at atmospheric $O_2$ concentration (21%). Oxygen partial pressure ($pO_2$) (mmHg) in each setting is represented in a color scale ranging from blue (lower) to red (higher). A $pO_2$ of <0.1 mmHg was considered anoxic (represented as white regions), as described for islets and acinar tissue[10,15–20]. The models depicted in Fig. 1b, c and Supplementary Movies 1, 2 predict that anoxia will appear in transwell-kept slices as early as 16 h after their placement in culture, with a precipitous drop in core oxygenation immediately afterwards (Fig. 1c). Additional estimations of the anoxic volume for a parametric sweep of OCRs are indicated in Fig. 1d. Some degree of anoxia is predicted to emerge in PFC-cultured HPSs at higher-than-average OCRs, but the relative volume of anoxic tissue is always significantly lower than that modeled for transwell-cultured controls. While we cannot exclude that other oxygen concentrations may result in even better outcomes, our models show that 21% is sufficient to prevent the formation of anoxic regions over 24 h. Therefore, for the sake of convenience, we kept an atmospheric $O_2$ concentration for all experiments.

We also confirmed experimentally the prediction that HIF-1α, which is upregulated in low oxygen concentrations, had significantly higher expression in transwell- vs. PFC-cultured slices after 24 h of culture (Fig. 1e).

### Metabolic differences in PFC-cultured vs. control HPSs.
The above mathematical predictions suggest that HPSs cultured in transwells may suffer the consequences of oxygen deprivation, amongst which a switch from oxidative phosphorylation to anaerobic glycolysis has been reported[21]. Since glycolysis is a less efficient means to generate energy (2 ATP/molecule of glucose vs. ~30 by oxidative phosphorylation), we further hypothesized that transwell-cultured HPSs would also exhibit a higher glucose consumption rate (GCR) compared to those cultured on PFC. To test these hypotheses, we proceeded to culture HPSs in transwells or PFC dishes ($n = 3$ independent donors). Medium aliquots were taken every 24 h (just before daily changing) for 10 days, and glucose consumption, lactate production and osmolarity measured on a Beckman Coulter Vi-CELL MetaFLEX (Brea, CA). Lactate production (an indirect measurement of glycolytic metabolism) was significantly higher in the transwell group vs. PFC, both overall (AUC $p$-value = 0.03) and at every time point analyzed (Fig. 1e). Also, as predicted, HPSs cultured in transwells displayed a higher glucose consumption rate than their PFC counterparts at every time point after day 1 (AUC $p$-value = 0.04) (Fig. 1f), an observation that likely reflects a heightened need of sugar to keep up with energy demands in partially anaerobic conditions.

We further measured ATP production for $n = 3$ slices cultured in either TW or PFC for 10 days. The average ATP production in transwell-cultured slices was 21.9 ± 8.3 pm/μg of protein, compared to 57.7 ± 42.5 pm/μg in those cultured in PFC, which is statistically insignificant ($p$-value = 0.06) (±, standard deviation, two-tailed $t$-test). However, the production of ATP (pm)/μg of glucose, normalized by μg of protein, was 38.5 in the PFC group vs. 6.8 in the TW group (a ~6-fold increase). Thus, while both groups have comparable production of ATP, slices cultured in PFC dishes generate it with six times less glucose. Figure 1g shows the higher relative uptake of the fluorescent D-glucose analogue 2-[N-(7-nitrobenz-2-oxa-1,3-diazol-4-yl)amino]-2-deoxy-D-glucose (2-NBDG) by slices cultured in transwells vs. those in PFC during a 10-minute incubation. Quantification of the mean fluorescence intensity (MFI)/DAPI⁺ cells in both groups (0.53 ± 0.15 for the transwells vs. 0.075 ± 0.06 in PFC, a 7-fold increase)

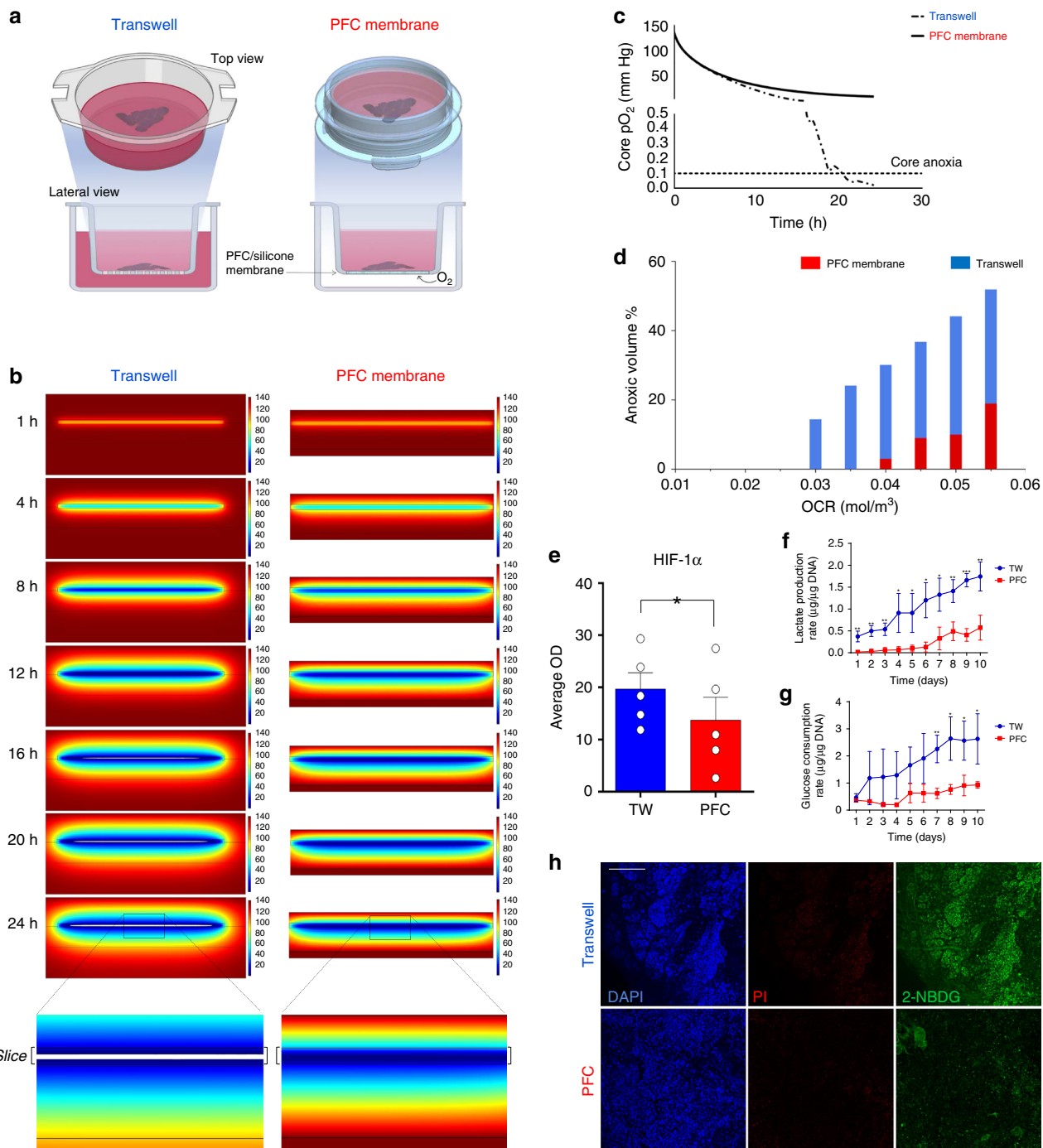

**Fig. 1 Enhanced oxygenation of human live pancreatic slices prevents anoxia and preserves oxidative phosphorylation. a** Diagram outlining the perfluorocarbon (PFC) membrane AirHive cell culture dish compared to a transwell dish. PFC dishes are selectively permeable to gases but not liquids. Therefore, medium can be added from the top while air is preferentially transferred directly from the bottom. Graphics partially done with BioRender. **b** COMSOL modeling of oxygen partial pressure across human pancreatic slices cultured in transwells (left) or PFC membranes (right) over a 24 h period. Spectral scale ranges from high (red) to low (blue) oxygen partial pressure. Anoxic areas are in white. **c** Partial pressure of oxygen within the core of the slice as a function of time. The dashed threshold represents anoxia. **d** Total anoxic volume of PFC- (red) and transwell- (blue) cultured slices modeled against increasing oxygen consumption rates. **e** HIF-1α protein level determination in transwell- vs. PFC- cultured slices after 24 h of culture ($n = 5$ biologically independent samples from individual donors). Two-tailed $t$-test: *$p$-value = 0.0304. **f** Lactate production rate of PFC- (red) and transwell- (blue) cultured slices as a function of time. $n = 3$ biologically independent samples from individual donors. Multiple two-tailed $t$-tests: **$p$-val$_{1day}$ = 0.0083, **$p$-val$_{2days}$ = 0.0037, **$p$-val$_{3days}$ = 0.0056, *$p$-val$_{4days}$ = 0.0319, *$p$-val$_{5days}$ = 0.0357, *$p$-val$_{6days}$ = 0.0111, *$p$-val$_{7days}$ = 0.0192, **$p$-val$_{8days}$ = 0.0092, ***$p$-val$_{9days}$ = 0.0004, ***$p$-val$_{10days}$ = 0.0009. **g** Glucose consumption rates of PFC- (red) and transwell- (blue) cultured slices as a function of time. $n = 3$ biologically independent samples from individual donors. Two-tailed $t$-tests: **$p$-val$_{7day}$ = 0.0063, *$p$-val$_{8days}$ = 0.0167, *$p$-val$_{9days}$ = 0.0244, *$p$-val$_{9days}$ = 0.0343. **h** Representative immunostaining of 2-NBDG uptake (green) during a 10-min incubation of human pancreatic slices cultured in transwells (top) or PFC dishes (bottom), reflecting the higher relative uptake of glucose of the former. Red: Propidium Iodide (PI), indicative of cell death. Blue: DAPI. Scale: 100 μm. **b**–**d** represent theoretical computational modeling and have no error bars. In **e**, data are presented as mean ± S.E.M; in **f**, **g**, as mean ± SD. Each $n$ further represents the mean of three technical replicates. Source are provided in the Source Data file.

confirms that the differences between the two groups are statistically significant (two-tailed $t$-test, $p$-value = 0.02). The lower glucose uptake in the PFC group was not due to lower viability of the tissue. In fact, the opposite is true, as there was a 2.6-fold increase in the ratio of propidium iodide (PI, cell death marker): DAPI in the transwell ($0.40 \pm 0.04$) vs. the PFC ($0.14 \pm 0.06$) group (two-tailed $t$-test, $p$-value = 0.007). This observation further supports the hypothesis that slices cultured in transwells require higher glucose uptake.

While carbon isotope techniques would be necessary to conclusively demonstrate metabolic changes, our data collectively suggest that culture of HPSs in PFC membranes preserves the usage of physiological oxidative phosphorylation as a primary source of energy over an extended period.

**Long-term viability of HPSs cultured on PFC-based membranes.** Decreased viability and tissue degradation are expected outcomes of sub-optimal oxygenation. We hypothesized that PFC-mediated enhanced oxygenation, coupled with proteolytic enzyme inhibition, would improve the overall long-term viability of HPSs vs. standard culture. To test this, we proceeded to culture HPSs from $n = 5$ independent (non-T1D) donors. (demographics in Supplementary Table 1). For each donor, $n = 20$ slices were assigned either to PFC ($n = 10$) or transwell ($n = 10$) groups. Consecutively cut slices were split between the two groups to minimize the impact of potential regional variations. Figure 2a shows live (green)/dead (red) imaging of a representative freshly sectioned slice, typically >80% viable (Fixed Viability Dye, Thermo-Fisher Scientific, Waltham, MA). A dead (fixed) slice is stained for comparison in Fig. 2b. Following extended culture, the viability (percentage of live/dead cells) (Fig. 2c–e, Supplementary Fig. 1b) and cross-sectional area (reflective of overall tissue integrity) of HPSs cultured in transwells are lower than those observed in PFC (Fig. 2f–g, Supplementary Fig. 1c). Slices from a T1D donor (Supplementary Table 1) also exhibit the same behavior when kept in either culture setting, as indicated in Fig. 2h–i. To further characterize the effects of culture conditions on HPSs, we performed immunofluorescence assays for endocrine and exocrine markers. Figure 2j shows rounded islets and well-preserved islet endocrine cell types both at day 0 and at day 10 of PFC culture. However, those of HPSs kept in transwells have less-defined boundaries (Fig. 2j).

All the quantified endocrine cell types were positive for their respective hormone and DAPI, suggesting nuclear integrity. The relative proportion of β-, α- and δ-cells remains unchanged (Fig. 2k) in all conditions. This observation, coupled with the overall reduction in cross-sectional surface (Fig. 2d, e) and smaller islet area in transwells at 10 days (Fig. 2l) suggests that progressive cell death in transwells occurs similarly across all cell populations. Otherwise, the proportion of endocrine cells vs. all the cells would be altered in transwells at day 10. Furthermore, the relative abundance of Ki67-positive cells is similar in both groups at day 10, with most proliferative cells observed in the exocrine compartment (Supplementary Fig. 2a). Therefore, culture in PFC dishes does not induce apparent changes in cell turnover throughout a 10-day period.

Examination of key β-cell transcription factors NKX6.1 and PDX1 at day 10 indicates that their expression is stronger in PFC than transwell conditions (Supplementary Fig. 2b, c). While no differences in amylase signal were apparent at low magnification between day 0 and day 10 on either condition (Fig. 2m), higher magnification images show that PFC-cultured slices have well-preserved E-cadherin ultrastructure and display the typical concentration of granules in the apical region of the cells (Supplementary Fig. 2d, bottom). In contrast, slices cultured for 10 days in transwells have less-defined E-cadherin and acinar morphology, with more diffuse amylase staining (Supplementary Fig. 2d, top). This resolution was insufficient to detect potential ultrastructural changes such as increase vacuole formation, basal membrane abnormalities or junctional complex disruption.

qRT-PCR was conducted to test whether the expression of key endocrine and non-endocrine genes was affected by long-term culture in PFC- vs. transwell-cultured slices. As shown in Supplementary Fig. 3, all the examined markers are upregulated in PFC vs. transwells, further suggesting that the former preserves better the integrity and function of HPSs.

**Enhanced oxygenation supports extended function in HPSs.** The claim that HPSs sustain function throughout long-term culture must be substantiated by functional assays for both the endocrine and exocrine compartments. To do so, we measured dynamic glucose-responsive insulin secretion (perifusion) and glucose-mediated calcium imaging (both for endocrine activity), as well as carbachol (CCh)-mediated release of digestive enzymes indicative of acinar function. Figure 3a shows the pattern of glucose-dependent insulin release (perifusion assay) in live slices shipped from nPOD's Gainesville laboratory (day +1$^S$, one day after sectioning, shipped; black), day +10 transwell (blue) and day +10 PFC (red) islet-containing HPSs from $n = 3$ independent donors. Perifusion charts for each donor are shown in Supplementary Fig. 4. Figure 3d presents the analysis of HPSs from a short-term T1D donor. Stimulation indices (SI) normalized to baseline are in the $Y$-axis, with areas under the curve (AUC) values in Fig. 3b, c (for non-diabetic donors) and Fig. 3e, f (for the T1D donor).

All three groups (day +1$^S$, day +10 transwell and day +10 PFC) exhibit glucose-stimulated responsiveness, although typical biphasic responses were not observed. However, in comparison with the PFC group, acute (i.e., shipped from nPOD) and transwell groups showed decreased insulin secretion at 16 mM glucose and 30 mM KCl stimulation. SIs were higher in PFC than in transwells, and differences were statistically significant (Fig. 3c).

Interestingly, acute samples (day +1$^S$) normally had poorer glucose responsiveness than those analyzed at day 10, which may reflect shipment stress. This mirrors the reduction of islet function immediately after isolation. In fact, islets are typically allowed to rest for 24–48 h prior to the conduct of functional assays, a period during which they deactivate stress signaling pathways[22,23]. To test whether resting the slices for 24 h after being subjected to stress would lead to functional recovery, we conducted additional experiments on $n = 3$ new donors. In this case, HPSs were generated in-house and perifused immediately after slicing (day 0), after 24 h recovery (day +1) and then at day +10 (transwells or PFC dishes). We hypothesized that day 0 HPSs would be stressed in the perisectioning period, thus having poorer responses than those allowed to rest an additional day. This hypothesis proved correct, as evidenced by a significant rebound in the SI of HPSs rested for 24 h (Fig. 3g–i). Of note, day 0 responses in this setting are roughly equivalent to those observed at day +1$^S$ in Fig. 3a, suggesting that both forms of stress (shipping and slicing) have a deleterious, yet transient effect on function. The glucose responsiveness of slices in PFC was higher than that recorded in the transwell group at day 10, but in this particular set of experiments (unlike in the previous one) the differences were not statistically significant. This is probably due to the high donor-to-donor variability, as shown in Supplementary Fig. 4b (individual donor perifusions) and Supplementary Fig. 3 (qRT-PCR). All the experiments subsequently presented showing acute measurements were conducted on rested slices.

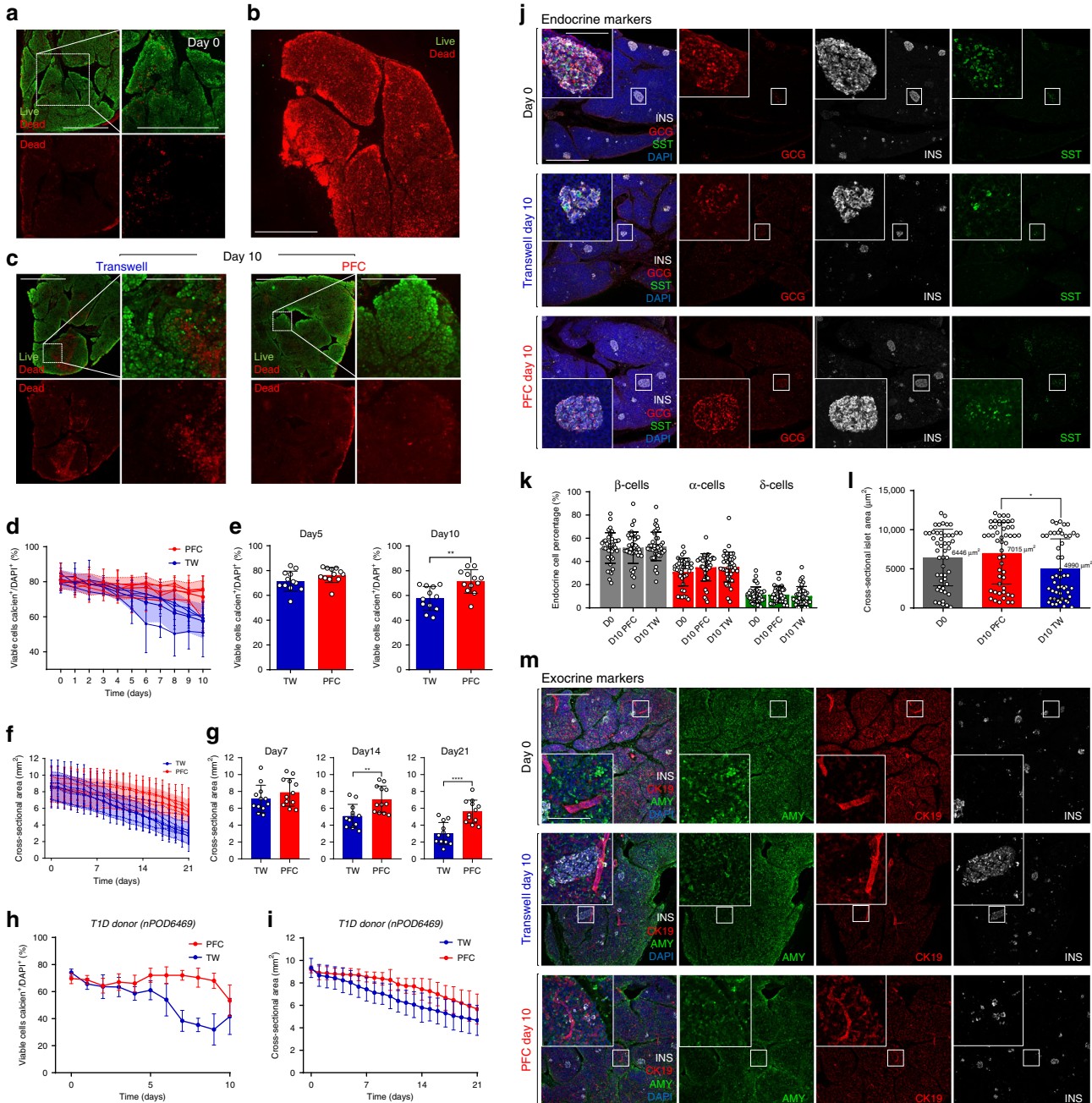

**Fig. 2 Human pancreatic slices remain viable after extended culture. a** Representative immunostaining showing tissue viability at the time of tissue slicing, showing live (green) and dead (red) regions. Scale: 500 µm for all panels. **b** Immunostaining of control fixed slice showing only dead cells. Scale: 500 µm. **c** Immunostaining showing slice viability assessments after 10 days of culture in either transwells or PFC dishes. Necrotic zones in transwell-cultured slices are highlighted in magnified panel. Overt necrosis, in contrast, is not apparent in PFC-cultured HPSs. Scale: 500 µm. Magnified insets: 200 µm. For **a–c**: $n = 5$ biologically independent samples from individual donors. **d** Quantitative measurement of viable cell percentages in PFC (red)- and transwell (blue)-cultured slices over a period of 10 days. Shaded area shows mean ± SD. **e** Bar plots showing cellular viability of either culture setting at day 5 and 10 post-slicing. Two-tailed $t$-test: **$p$-val$_{10days}$ = 0.002. **f** Slice cross-sectional area of slices kept in either culture condition for 10 days in culture. Shaded area shows mean ± SD. **g** Individual bar plots showing slice cross-sectional area at days 7, 14, and 21 post slicing. Two-tailed $t$-test: **$p$-val$_{14days}$ = 0.0036, ****$p$-val$_{21days}$ = 0.000087. For **d–g**: $n = 4$ biologically independent samples from individual donors. **h** Viable cell percentages of PFC- and transwell-cultured T1D donor slices over a period of 10 days ($n = 1$ sample from an individual donor). **i** Slice cross-sectional area in either culture condition over 10 days of culture (T1D donor) ($n = 1$ sample from an individual donor). **j** Immunofluorescence images showing expression of representative endocrine markers (INS insulin, white; GCG glucagon, red; and SST somatostatin, green) within islets of slices at day 0 and cultured for 10 days in either transwells or PFC dishes. Scale for all panels: 500 µm; 100 µm for insets. **k** Dot plot showing the distribution of endocrine cells in islets at day 0 and after 10 days of culture (PFC and transwells). **l** Dot plot showing the cross-sectional area of islets at day 0 and after 10 days of culture (PFC and transwells). Two-tailed $t$-test: *$p$-val$_{D10 (PFC)-D10 (TW)}$ = 0.0107. For **j–l**: data from $n = 10$ biologically independent samples from individual donors. **m** Immunostaining showing insulin (INS, white) and representative exocrine markers (cytokeratin-19, CK-19, red; and amylase, AMY, green) within slices at day 0 and after 10 days of culture (PFC and transwells). Blue, DAPI counter-staining. $n = 10$ biologically independent samples from individual donors. Scale: 500 µm; 100 µm for insets. For **d–i**, **k**, and **l**, data are presented as mean ± SD. Each $n$ further represents the mean of three technical replicates. * $p < 0.05$; **$p < 0.01$; ***$p < 0.005$. Source are provided in the Source Data file.

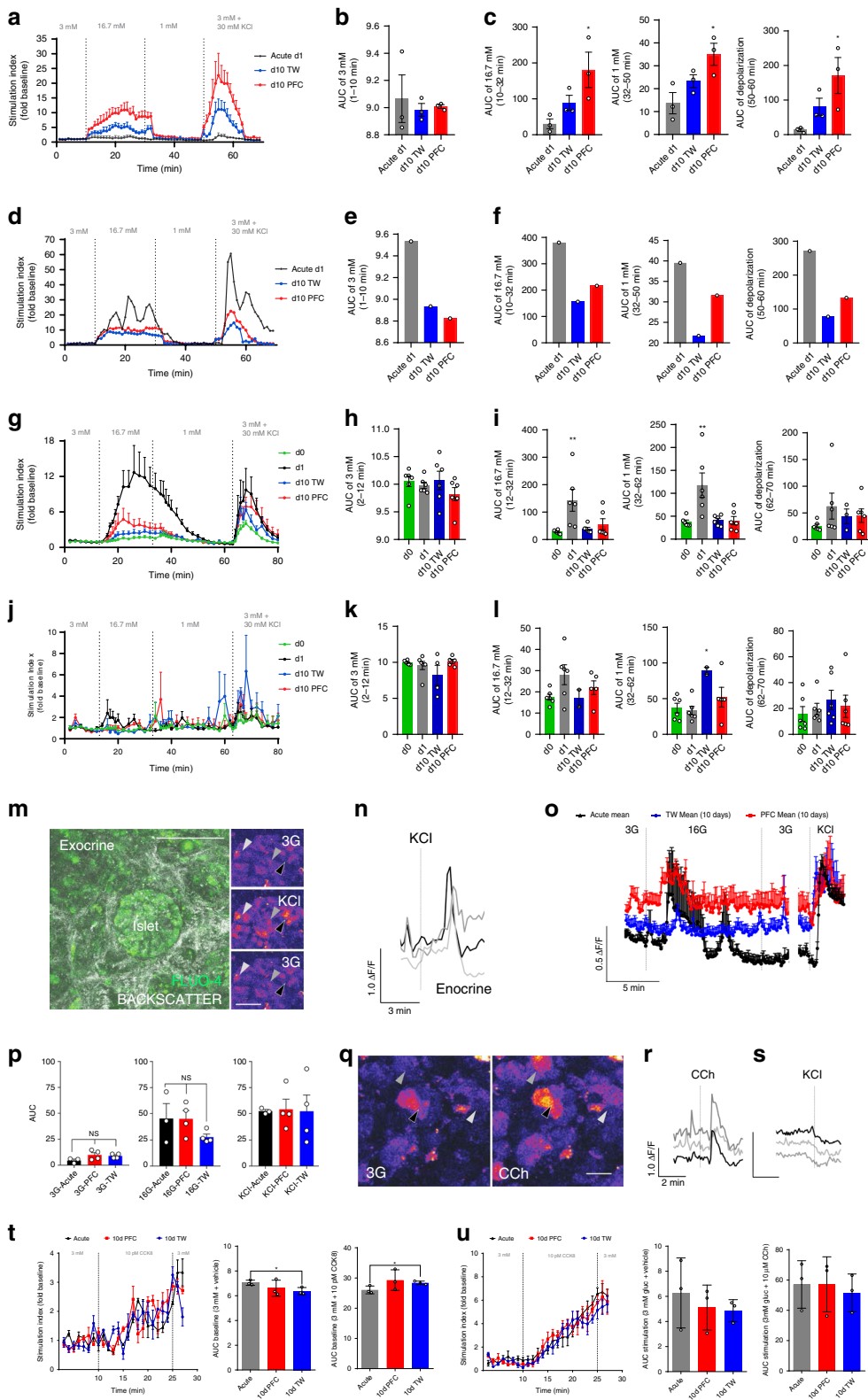

Figure 3j–l also shows dynamic glucagon secretion with glucose stimulation. Although glucagon responsiveness is more difficult to measure in HPSs due to the smaller number of alpha cells in each sample, the rapid peak detected immediately after 1 mM (low) glucose in the PFC group (Fig. 3j, red trace) is physiological, unlike the delayed one observed in long-term transwell-cultured HPSs (Fig. 3j, blue trace).

As $Ca^{2+}$ imaging reflects β-cell activity[24], we set out to determine whether HPSs cultured for 10 days in either condition exhibited changes in cytoplasmic $Ca^{2+}$ concentration in response to glucose stimulation, using the calcium indicator Fluo-4AM. This technique allows to measure β-cell stimulation in response to secretagogues such as 16.7 mM glucose or 30 mM KCl (Fig. 3m, n and Supplementary Movie 3). Select β-cells were observed to

**Fig. 3 HPSs retain function after extended culture. a** Perifusion of HPSs after high glucose (16.7 mM, 16 G) and KCl (30 mM) stimulation. X-axis: time/[Glucose]; Y-axis: SI Stimulation Index. Black: acute, day $+1^S$; red: PFC, day 10; blue: day 10 transwell. **b** Area under the curve (AUC) at 1–10 min (3 mM glucose/baseline) ($n = 3$). **c** Left to right, AUC at 10–32 min (16.7 mM, high glucose); 32–50 min (1 mM, low glucose); and 50–60 min (KCl depolarization). **a**–**c**: $n = 3$ biologically independent samples from individual donors). One-way ANOVA post hoc Tukey HSD: $*p\text{-val}_{16.7\text{mM[D1 (acute)-D10 (PFC)]}} = 0.0362$, $*p\text{-val}_{1\text{mM[D1 (acute)-D10 (PFC)]}} = 0.0266$, $*p\text{-val}_{\text{Depolarization[D1 (acute)-D10 (PFC)]}} = 0.0368$. **d** Perifusion of T1D HPSs after high glucose and KCl stimulation. Y-axis, SI. **e** AUC at 1–10 min (3 mM glucose). **f** Left to right, AUC as in **c** ($n = 1$ sample from an individual T1D donor). **g** Perifusion of HPSs from additional donors after high glucose and KCl stimulation. Traces show responses at day 0 (immediately after slicing; green), d $+1$ (after a 24 h rest; black) as well as day $+10$ in transwells (blue) and PFC (red). Y-axis, SI. **h** AUC at 1–10 min (3 mM glucose). **i** Left to right, AUC as in **c**. One-way ANOVA post hoc Tukey HSD: $**p\text{-val}_{16.7\text{mM[d0-d1]}} = 0.0047$, $*p\text{-val}_{1\text{mM[d0-d1]}} = 0.0023$. **g**–**i**: $n = 3$ biologically independent samples from individual donors. **j** Glucagon perifusion of HPSs after stimulation with high glucose and KCl. Y-axis, SI. **k** AUC at 1–10 min (3 mM glucose). **l** Left to right, AUC as in **c**. **j**–**l**: $n = 3$ biologically independent samples from individual donors. **m** Representative microphotograph of islet (backscatter) in a PFC-cultured slice at day 10, loaded with Fluo-4-AM. Right panels: stimulation with 30 mM KCl. Arrows point at three endocrine cells. **n** Traces of $Ca^{2+}$ responses in cells after KCl stimulation, color-coded to those indicated in **m**. **o** Trace of β-cells responding to high glucose and KCl stimulation. **p** AUC of $Ca^{2+}$ responses after high glucose and KCl stimulation. **m**–**p**: $n = 3$ biologically independent samples from individual donors. **q** Representative microphotograph of exocrine tissue upon stimulation with carbachol (10 μM). Arrows point at three acinar cells. **r** Color-coded traces for carbachol stimulation of the cells in **q**. **s** Traces of cells exposed to KCl after responding to carbachol, indicating non-endocrine phenotypes. **t** Left, amylase secretion after 10 pM CCK8 stimulation. Middle, AUC for 3 mM glucose/baseline stimulation. Right, AUC for 10 pM CCK8. **q**–**t**: $n =$ three biologically independent samples from individual donors. Two-tailed t-test: $*p\text{-val}_{3\text{mM[Acute-d10TW]}} = 0.0307$, $****p\text{-val}_{21\text{days}} = 0.0450$. **u** Left, amylase secretion after 10 μM carbachol stimulation. Middle, AUC for 3 mM glucose/baseline. Right, AUC for 10 μM carbachol ($n = 3$). For **a**–**c**, **g**–**l**, and **o** data are presented as mean ± SEM, while for **p**, **t**, **u** data are presented as mean ± SD. $n = 1$ chamber in **d**–**f**, hence no error bars. Each $n$ further represents the mean of three technical replicates (three slices in one chamber in case of **a**–**l**, **t**–**u**). $*p < 0.05$; $**p < 0.01$; $***p < 0.005$. Source data are provided in the Source Data file.

respond (increased cytoplasmic $Ca^{2+}$) to 16.7 mM glucose as well as 30 mM KCl in HPSs cultured for 10 days atop both transwells and PFC membranes. KCl responses were not significantly different between day 10 PFC- and transwell-cultured HPSs. However, only PFC-cultured HPSs showed a significant difference between baseline (3 mM) and stimulation concentrations (16.7 mM) of glucose (Fig. 3o–p). We found β-cells to be synchronous and oscillatory in both groups, without disruption in coupling. Overall, pulses in cultured transwell-cultured slices during 16.7 mM glucose stimulation were weaker than those in PFC, but the results were not significantly different.

Exocrine cells respond to acinar-specific secretagogues such as carbachol (acetylcholine agonist) and cholecystokinin[25]. In this respect, we observed that exocrine cells in HPSs responded to both (Fig. 3q–u, Supplementary Movie 4) without depolarization by KCl (Fig. 3s), which is typical of endocrine cells[26]. Acinar activity was also synchronized during carbachol stimulation.

**Long-term HPS culture enables β-cell regeneration studies**. To establish proof of principle that extended culture of HPSs could be used to study regeneration, we generated slices from a transgenic mouse (INS$^{Cre}$mTmG)[14] where insulin-producing cells are labeled in green (EGFP), whereas the remaining cells are red (mTomato) (Fig. 4a). This mouse has been used to study whether β-cell neogenesis occurs from pre-existing β-cells or from non-β-cells[14]. In this model, if new β-cells arise from pre-existing ones, they will remain green. However, when a non-β-cell (e.g., a progenitor, red) begins to express the insulin promoter, a Cre-mediated recombination event leads to the excision of mTomato and the expression of EGFP. There is a window during which the mTomato protein has not been degraded yet but EGFP is already expressed. Hence, the transient yellow (red + green) color of these cells can be used to identify β-cell neogenesis[14].

BMP receptor agonists have been shown by our group to induce the activation of progenitor-like cells residing in the major ductal tree of the human pancreas[27,28]. Mirroring the experimental design previously used with human non-endocrine pancreatic tissue[27] as well as sorted progenitor-like cells[28], we hypothesized that stimulation with a BMP receptor agonist, followed by withdrawal thereof, would result in detectable β-cell neogenesis in pancreatic slices. If new β-cells were to appear from non-β-cells (e.g., progenitors), we would observe the conversion

of red into green cells. To test this hypothesis, slices were generated from 6-8-week-old INS$^{Cre}$mTmG mice, placed in PFC dishes as previously described for HPSs, and treated for 5 days with 100 ng/ml of BMP-7. Control slices received vehicle instead of BMP-7. From day 6–9, BMP-7 was no longer administered. As shown in Fig. 4b, newly formed insulin$^+$ cells were observed starting at day 9 in regions that had been previously devoid of green (insulin) signal. No such occurrence was detected in controls. Figure 4c presents another similarly designed experiment using a BMP-7-like agonist, THR-123[27,29]. In this case, green cells were detected from day 7, mostly in a region corresponding to a large pancreatic duct.

To see whether we could replicate this model in non-transgenic mice, we co-transduced pancreatic slices from CD-1 (wild-type) mice with adenoviruses carrying the reporter construct CMV-loxP-dsRED-loxP-EGFP (kindly given by P. Ravassard) and a rat insulin promoter (RIP)-driven Cre recombinase (Fig. 4d). We predicted that doubly transduced non-β-cells would be tagged red and pre-existing β-cells would be tagged green. As in the previous transgenic setting, an additional prediction was that doubly transduced insulin$^+$ cells arising from non-β-cells would be transiently tagged red with decaying dsRED and permanently green with newly produced EGFP (transient yellow). We treated slices with BMP-7 as above. In all, 24 h after the double transduction, islets were labeled green, as shown in the lower-right inset of Fig. 4e. The formation of new insulin-expressing cells from non-β-cells was confirmed in several regions of the slices, normally associated to ductal areas, where groups of red cells progressively transitioned to green following the treatment with and subsequent withdrawal of BMP-7 (Fig. 4e). Red and green signal could be readily monitored and quantified as a function of total slice area longitudinally. When this experiment was repeated using slices treated with the β-cell toxin alloxan, control (non-BMP-7-treated) slices displayed virtually no green signal throughout (Fig. 4f–h). As an additional internal control, non-alloxan-treated slices (dashed-green line, Fig. 4h) exhibit readily detectable green signal (corresponding to islets). The red signal, corresponding to reporter-transduced slices, was similar for all the experiments (Fig. 4g), indicating no aberrant transduction efficiency due to alloxan treatment. However, when the slices were treated with BMP-7, green signal was progressively detected after alloxan administration, demonstrating new insulin$^+$ cell formation (Fig. 4i–k). Partially confirming the β-cell

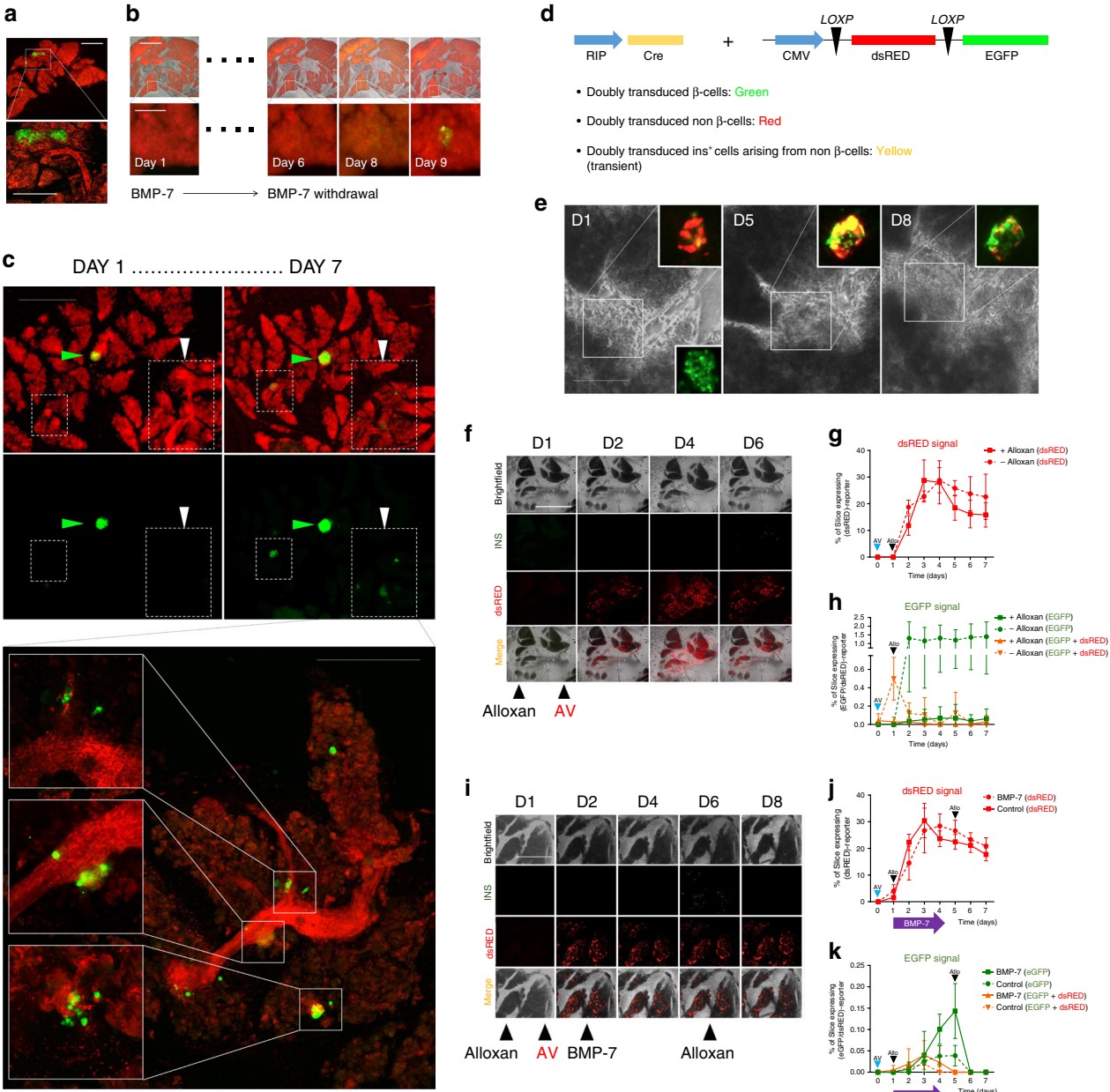

**Fig. 4 Tracking of β-cell regeneration in murine pancreatic slices. a** Representative microphotograph of murine pancreatic slices from Ins2-Cre.mTmG mice, with EGFP+ insulin-producing cells (green) and all other cells tdTomato+ (red). An islet and adjacent duct are magnified in the inset. Scale 300 μm. **b** Following 5 days of BMP-7 stimulation, removal thereof leads to insulin+ cell formation at d9. Scale 300 μm (magnified insets: 50 μm). **c** Representative microphotograph of murine slices from Ins2-Cre.mTmG mice treated for 7 days with THR-123. Green arrow shows a pre-existing islet. After THR-123 removal, new insulin+ cells appear across the slice. Higher magnification confocal imaging of the same region (lower panel) shows new insulin+ cells sprouting from ducts. Scale 200 μm; magnified panel: 100 μm. **d** Reporter strategy involving the co-transduction with adenoviral RIP-Cre tracer and lox/dsRED/lox-EGFP reporter. Below are predicted outcomes depending on cell identity. **e** Representative microphotograph of CD1 mice-derived slices co-transduced as in **d**. BMP-7 is added for 5 days and then withdrawn. D1-d8 images show progressive appearance of insulin-expressing cells (green) from insulin− cells through an intermediate yellow (red + green) stage. Pre-existing islet-resident β-cells are EGFP-tagged within the first 48 h of transduction, as shown in d1 inset. Scale 50 μm. **f** Longitudinal imaging of CD1 mice-derived pancreatic slices transduced as in **d**. Alloxan was used to ablate β-cells at d1. No BMP-7 was added. Scale 500 μm. **g** Expression of dsRED protein in transduced control slices over 7 days in alloxan-treated (+) and untreated (−) slices. **h** EGFP expression and co-localization of EGFP + dsRED in co-transduced alloxan-treated (+) and untreated (−) slices over 7 days. **i** Representative microphotograph of CD1 mice-derived slices co-transduced as in **d**. Slices were also treated with alloxan to ablate β-cells and subsequently exposed to BMP-7 from day 2 to day 4. Scale 500 μm. **j** Quantification of dsRED expression in co-transduced, BMP-7-treated or control slices over 7 days in alloxan-exposed slices, showing no differences between groups. **k** Quantification of EGFP and co-localization of EGFP + dsRED (yellow) in co-transduced alloxan-exposed slices following treatment with BMP-7 (solid lines) or vehicle (dotted lines). Slices were challenged with alloxan a second time at d5, ablating new insulin+ cells. Scale 500 μm. For **a**–**k**, n = 5 biologically independent samples from individual mice. Data are presented as mean ± SD. Each n further represents the mean of three technical replicates, while plotted bars/lines are centered at mean. Source data are provided in the Source Data file.

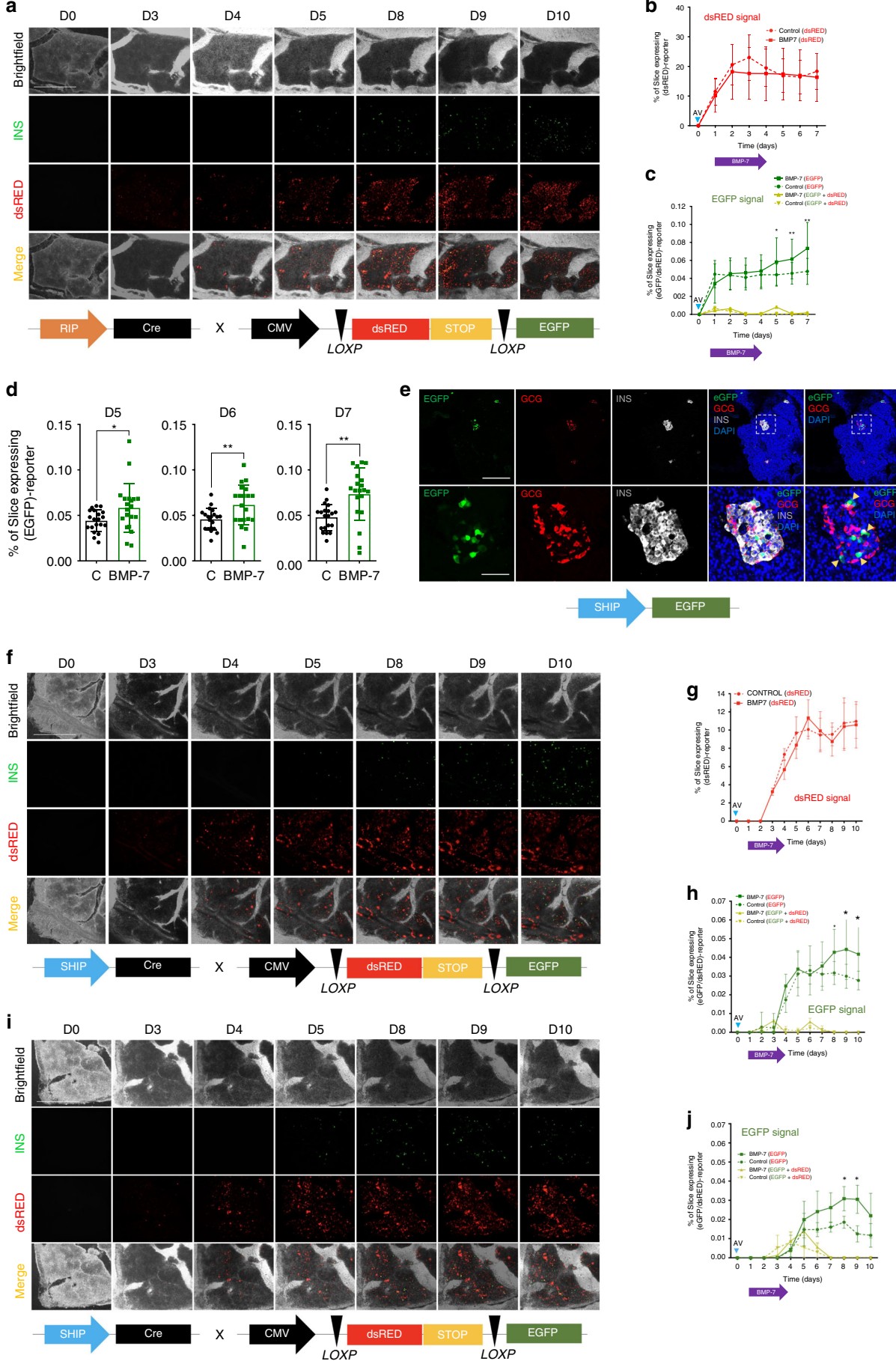

**Fig. 5 Longitudinal tracking of β-cell formation in HPSs over extended culture. a** Longitudinal red-green immunofluorescence imaging of HPSs co-transduced with RIP-Cre and reporter adenoviruses (viral constructs in outline below). Scale 500 μm. **b** Quantification of dsRED expression in co-transduced BMP-7- (solid red line) or vehicle- (dotted red line) treated slices over time, showing no significant differences between groups. **c** Quantification of EGFP and co-localization of EGFP + dsRED (yellow) in co-transduced HPSs following treatment with BMP-7 (solid lines) or vehicle (dotted lines). Two-tailed t-test: *p-val$_{5 \text{ days}}$ = 0.0363, **p-val$_{6\text{days}}$ = 0.0076, **p-val$_{7\text{days}}$ = 0.001. **d** Percentage of slice area expressing EGFP at d5, 6 and 7, as shown in **c**. Two-tailed t-test: *p-val$_{5\text{days}}$ = 0.0363, **p-val$_{6\text{days}}$ = 0.0076, **p-val$_{6\text{days}}$ = 0.0011. **e** Representative confocal immunofluorescence imaging of a region of a slice containing islet-resident β-cells tagged by a SHIP-EGFP adenovirus. HPSs were transduced with the SHIP-EGFP adenovirus, and slices were then fixed for immunostaining after 30 h. The lower row shows a magnified region for each channel of an islet with untagged α-cells (GCG, Glucagon+), whereas some β-cells (insulin+) are tagged by SHIP-EGFP. Scale, top row, 200 μm; bottom, 50 μm. **f** Longitudinal immunofluorescence imaging of HPSs co-transduced with SHIP-Cre and reporter adenoviruses (viral constructs in outline below). Scale 500 μm. **g** Quantification of dsRED expression in co-transduced BMP-7- (solid red line) or vehicle- (dotted red line) treated slices over 10 days in human pancreatic slices, showing no significant differences between groups. **h** Quantification of EGFP (green) and co-localization of EGFP + dsRED (yellow) in co-transduced human slices following treatment with BMP-7 (solid lines) or vehicle (dotted lines) over 10 days. For **a–h**, n = 3 biologically independent samples from individual donors were used. Two-tailed t-test: *p-val$_{8\text{days}}$ = 0.0278, **p-val$_{9\text{days}}$ = 0.0237, **p-val$_{10\text{days}}$ = 0.0144. **i** Longitudinal red-green immunofluorescence imaging of HPSs from a T2D donor co-transduced with SHIP-Cre and reporter adenoviruses (viral constructs in outline below panel) (n = 1 sample from an individual donor). Scale 500 μm. **j** Quantification of EGFP and co-localization of EGFP + dsRED (yellow) in co-transduced HPSs following treatment with BMP-7 (solid lines) or vehicle (dotted lines). The quantification of dsRED expression in co-transduced BMP-7- or vehicle-treated slices indicated no significant differences between the two groups (n = 1). Two-tailed t-test: *p-val$_{8\text{days}}$ = 0.039, **p-val$_{9\text{days}}$ = 0.02. For **b–d**, **g**, **h**, and **j** data are presented as mean ± SD. Each n further represents the mean of three technical replicates, while plotted bars/lines centered at mean. *p < 0.05; **p < 0.01; ***p < 0.005. Source data are provided in the Source Data file.

identity of the green cells, when alloxan was added again at this point, all EGFP+ cells disappeared within 24 h (Fig. 4k).

Based on these results in mice, we sought to replicate this adenoviral-based reporter system using HPSs. Slices were generated from n = 5 donors and treated one day after transduction with 100 ng/ml BMP-7 for 3 days, followed by removal for 3 days (Fig. 5a–d). Since human β-cells are naturally resistant to alloxan, pre-existing islets could not be eliminated as previously done in murine slices. Therefore, co-transduction of the two adenoviral vectors was expected to result in the labeling of resident β-cells, whereas newly formed β-cells would represent a comparatively smaller contribution to the overall EGFP signal. Another prediction of the model was that, in the absence of additional regeneration (i.e., no BMP-7 control), EGFP signal corresponding exclusively to islets would plateau soon after all transduced β-cells turned green. In contrast, BMP-7-mediated formation of new insulin-producing cells would keep gradually increasing in BMP-7-treated slices after controls have plateaued. Confirming these hypotheses, EGFP signal in the BMP-7-treated group continued to expand with time, whereas the islet-labeled area in the control group remained unchanged after day 1 (Fig. 5c, d).

Since the insulin promoter used was not human, we set out to confirm our findings using a synthetic human insulin promoter (SHIP) with heightened specificity for human β-cells[30] (kindly provided by Dr. C. Brunicardi, University of Toledo, OH). Using an adenovirus containing a SHIP-driven EGFP construct, we tested for promoter specificity. As expected, we observed that all SHIP-labeled cells were insulin+ (Fig. 5e and Supplementary Fig. 5a, c, d), but not glucagon+ (Fig. 5e and Supplementary Fig. 5d), somatostatin+ (Supplementary Fig. 5b, d) or detected outside islets (Supplementary Fig. 5e). Next, we co-transduced HPSs (n = 3 donors), treated them with BMP-7 (or vehicle) for 3 days and tracked β-cell formation over time. As before, following the initial labeling of pre-existing islets in both groups (which is delayed vs. that observed with RIP-Cre), we observed a progressive increase of the EGFP-tagged area in the BMP-7-treated group, whereas that of the control group remained largely unchanged. The quantification of this signal showed significant differences in total EGFP-expressing area at days 8–10 (4–6 days after BMP-7 removal), confirming β-cell generation using a human specific promoter system (Fig. 5g, h). Both with RIP and SHIP, waves of yellow-tagged cells (Fig. 5c, h) were more readily detected in BMP-7-treated slices than in controls.

Finally, we repeated the experiment using HPSs from a T2D diabetic donor. While rarer, these samples present us with the unique opportunity to study regeneration in a disease model. As shown in Fig. 5i, j, these samples also exhibited BMP-7-dependent increase of EGFP signal from day 8 onwards vs. controls, suggesting that β-cells can be induced to regenerate in slices from T2D donors.

## Discussion

We describe here the long-term culture of HPSs using oxygen-enhancing techniques. These methods resulted in the functional preservation of endocrine and exocrine cells for at least 10 days, a period during which HPSs cultured in control conditions exhibited significant deterioration. The extended preservation of organotypic pancreatic tissue, coupled with adenoviral-mediated lineage tracing, offers a unique opportunity to explore human endocrine regeneration in a culture setting that retains to a significant degree the anatomical integrity of the native organ. While previous reports have lineage-traced human primary pancreatic tissue in vitro[27,31–33], the very act of placing such tissues in culture introduces experimental bias and induces artifactual de-differentiation[27,32,33]. HPSs have circumvented those problems[3–7], but their usefulness has been limited to short-term physiological studies due to their rapid degeneration. To our knowledge, no other report on organotypic human pancreatic tissues describes culture for longer than 48 h.

Suboptimal oxygenation is arguably the most critical factor responsible for this loss of viability and function. The thickness of HPSs (120 μm) is at the very limit of passive gas diffusion across living tissue that would ensure survival of cells at its core, provided that an adequate oxygen supply (e.g., a blood vessel) existed at both sides. This is not the case in regular settings, where air has to diffuse through a much longer distance from the surface of the medium to where the tissue is located—and then across it. Our modeling confirms that anoxia occurs rapidly after only 16 h of culture in transwells. Anoxia induces an irreversible loss of respiratory capacity in the pancreas[34]. This leads not only to cell death but, perhaps more importantly, a metabolic change in the surviving cells, which adapt to lower environmental oxygen by switching to glycolysis. Even though they still represent an improvement over non-organotypic cultures, HPSs in transwells are metabolically aberrant. In contrast, PFC-cultured HPSs maintain low levels of lactate production and glucose

consumption rates after 10 days in culture. Since slices in both groups produce comparable amounts of ATP, but the glucose uptake and consumption are lower in those cultured in PFC, our observations suggest the maintenance of a highly efficient, oxidative phosphorylation-based metabolism. Additionally, PFC-cultured HPSs displayed significantly better viability, cross-sectional area, glucose responsiveness, and overall islet integrity over the studied period. We speculate that this is due to the suboptimal wellbeing of islets in control conditions[35]. Interestingly, those improvements were more evident in the endocrine than in the exocrine compartment. With the exception of the cross-sectional area, which was significantly higher in the PFC group, calcium imaging following exposure to acinar cell secretagogues and overall IF profiles were largely comparable in both groups at day 10. This may be due to refinements in the medium, which contains two additional protease inhibitors (aprotinin and chymostatin) compared to the formulations reported thus far for HPSs[1,9]. Still, higher magnification images of the acinar compartment indicate more defined E-cadherin ultrastructure and apical amylase granulation in PFC-cultured slices.

Another intriguing observation is that the acute dynamic glucose response profile of HPSs immediately after sectioning was worse than that recorded at day 10, be it on transwells or PFC dishes. This prompted us to test whether resting the samples for 24 h after the stress of slicing would result in their functional recovery, as it happens with islets after isolation[22,23]. This was indeed the case, as shown in the second round of experiments presented in Fig. 3g.

Our goal in refining the conditions for the long-term survival of HPS was to allow for the real-time detection and quantification of endocrine cell regeneration. Multiple models have been presented in the literature to explain β-cell regeneration, from β-cell duplication[36] and α-to-β cell transdifferentiation[37,38] to ductal progenitor cell differentiation (reviewed in ref. [39]). We decided to test the latter (ductal progenitors) owing to our previous discovery of a BMP-7-responsive sub-population of progenitor-like cells in the major ductal tree of the human pancreas[27,28]. However, our system could as easily be tailored to study other models of regeneration[37,38,40,41].

Proof-of-concept experiments were performed on a transgenic mouse model that was first used to study β-cell neogenesis in adult mice[14]. That report confirmed neogenesis during pancreatic development and in newborn mice, but failed to detect yellow (neogenic β-) cells after post-natal day 5 following a variety of insults to the pancreas. However, we readily detected yellow cells in slices derived from these very same mice after treatment with BMP-7 and THR-123. This apparent contradiction is not such, inasmuch as BMP-induced regeneration was not among the models tested in that study. Notably, we were able to replicate this in slices from non-transgenic mice, and subsequently from humans, by means of adenoviral co-transduction of a red-green reporter and an insulin lineage tracer. As predicted, the combination of long-term slice culture and virally-mediated lineage tracing allowed us not only to detect, but to quantify BMP-7-induced regeneration events longitudinally. The relatively low frequency of co-transduction was not an obstacle to observe transitional yellow and subsequently permanent green events in HPSs, both with the rat insulin promoter (RIP) and with a human synthetic insulin promoter (SHIP). Of note, owing to its molecular design, the SHIP promoter could be potentially activated in PDX1+ cells that do not express insulin[30]. However, we did not observe such occurrence upon transduction of human pancreatic slices with SHIP-Cre (Fig. 5e and Supplementary Fig. 5). Although the relative percentage of green events was lower when using the latter (suggesting a higher specificity of this promoter), the differences were not statistically significant.

The increase in EGFP population is statistically significant in the BMP-7 group vs. control. While the increase in the "yellow" population (i.e., cells that are transiently both EGFP- and dsRED-positive, reflecting new onset of insulin expression) is not statistically significant, it is important to stress that, while green signal is cumulative, yellow is not. Unlike the green signal, the yellow that we detect at the time of imaging may or may not be representative of what happens over a 24 h period. Therefore, determining true statistical significance based on the arbitrary temporal measurement of cells that are yellow only for a short period may not be possible.

An observation that stands out from the murine slice experiments is that new green cells are also detected in controls (non-BMP-7-treated), although at a significantly lower level. Both groups were treated with a β-cell toxin (alloxan), which in itself is a stimulus for regeneration[42]. This effect is likely aided by the high concentration of BMPs present in the serum used to supplement the murine (but not the human) culture medium[43]. In HPSs, we did not detect any increase of EGFP signal in controls after the initial tagging of pre-existing β-cells. In contrast, BMP-7-treated slices displayed a time-dependent upward trend in EGFP signal and significantly stronger waves of red-green co-localization, suggesting additional formation of β-cells after the initial labeling of the resident ones.

Importantly, our analyses are merely suggestive of β-cell neogenesis. The unequivocal demonstration of this phenomenon would require additional lineage-tracing as well as endocrine cell turnover determinations. Future work on such approaches will shed additional light on these observations. In addition, owing to the use of longitudinal slice-wide, low-magnification imaging (which is necessary to address the uncertainty about the location in the slice where new insulin+ cells may appear), transitional events cannot be resolved at the single cell level. However, while higher-resolution alternatives are developed, our analyses are still valid to quantify overall fluorescence changes at the tissue level. It must also be stressed that the in-depth mechanistic analysis of the reported observations (e.g., the functional analysis of newly formed insulin-producing cells) is beyond the scope of this report, whose main focus is the description of a method for the long-term, real-time monitoring of regeneration in live pancreatic cultures that preserve the tissue cytoarchitecture and cell-to-cell interactions of the native organ. This approach fills a void in the field and will enable investigators to study regeneration in the human pancreas with an unparalleled degree of resolution. If applied to samples from T1D and T2D donors, long-term cultured HPSs may offer new insights on the pathology—and potential treatment—of diabetes.

## Methods

**COMSOL modeling-finite element modeling of oxygen distribution**. 2D diffusion reaction modeling of oxygen distribution within the pancreatic slices was performed using COMSOL v5.3 (Comsol Inc, Burlington, MA)[10,11]. A parametric sweep of oxygen consumption rates ($1.0–5.0 \times 10^{-2}$ mol/m³ s⁻¹) unique to pancreatic tissue was performed[15]. The only conditions that differed between the two culture models were the basal surface boundary condition with continuity utilized for the transwell suspended in culture medium above the impermeable plastic surface, and a concentration boundary at the basal surface of the oxygen sandwich representative of the interface of the surface with the surrounding bulk oxygen. All diffusivity was represented as diffusive permeability, the product of effective diffusivity and the oxygen solubility in a given phase, in order to eliminate the need for partition coefficients and to express the oxygen distribution in terms of partial pressure (mmHg). Tissue anoxia was represented by a $pO_2$ of <0.1 mmHg per literature convention in regards to pancreatic islets and acinar tissue[10,15–20]. Anoxic tissue volume was calculated as the ratio of pancreatic slice volume with a $pO_2 < 0.1$ mmHg to total pancreatic slice volume, assuming each slice was a 1–1.2 cm square.

**Metabolic measurements**. Medium samples were collected from each transwell or PFC dish daily after the second medium change (12 h culture period) for the

measurement of metabolic activity. Media analytes were measured on a Beckman Coulter Vi-CELL MetaFLEX to determine glucose consumption, lactate production and osmolarity of the media samples. At every measurement timepoint, a medium blank was performed to normalize all pancreatic slice samples. In culture samples, evaporation is a confounding factor, so all values were corrected using the relative osmolarity of the culture sample relative to the media blank expressed in the following equations for glucose consumption and lactate production:

$$ \text{GCR} = \frac{G_{\text{blank}} - \left(G_{\text{culture}} \times \frac{\text{Osm}_{\text{blank}}}{\text{Osm}_{\text{culture}}}\right)}{\Delta\text{Time}} $$

$$ \text{LPR} = \frac{\left(L_{\text{culture}} \times \frac{\text{Osm}_{\text{blank}}}{\text{Osm}_{\text{culture}}}\right) - L_{\text{blank}}}{\Delta\text{Time}} $$

All values for GCR were expressed as (µg consumed/produced per hour)/µg DNA, while values for LPR were expressed as (µg produced/produced per hour)/µg DNA.

Glucose uptake was measured as follows: slices were pre-incubated in glucose-free BrainPhys medium (StemCell Technologies, Vancouver, BC, Cat# 05790) containing 2% B27 supplement minus-insulin (Invitrogen, Carlsbad, CA; Cat# A1895601) for 40 min. After washing 3X in PBS, slices from each group (PFC or TW) were incubated for 10 min in regular BrainPhys medium (see composition below) supplemented with 100 µM of 2-NBDG (Stemcell Technologies, Vancouver, BC, Cat# 05790). After incubation, slices were immediately washed 10X in PBS; briefly incubated with propidium iodide (PI), washed again 3×; briefly incubated with DAPI, washed again 3×, and immediately imaged using a confocal microscope.

**DNA quantification of slice tissue**. Pancreatic slices were digested using T-PER Tissue protein extraction reagent (Invitrogen, Carlsbad, CA, Cat# 78510). Supernatant was then used to calculate total DNA quantity using a Quant-iT Picogreen dsDNA assay kit (Invitrogen, Carlsbad, CA, Cat# P7589) according to the manufacturer's recommendations.

**Transgenic mouse models**. All animal experiments were conducted under the supervision and oversight of the University of Miami Institutional Animal Care and Use Committee (IACUC) and Division of Veterinary Resources (DVR) at the University of Miami. CD1-IGS mice (5–6 weeks old; Charles River, Wilmington, MA, Cat# 022) were utilized for pancreatic tissue slicing to obtain control slices for adenoviral transduction experiments. In order to create the INS2-Cre/mTmG reporter, we crossed *B6.Cg-Tg(Ins2-Cre)25Mgn/J* (INS2-Cre; Jackson Labs, Bar Harbor, ME, Cat# 003573) with *B6.Gt(ROSA)26Sor^{tm4(ACTB-tdTomato,-EGFP)Luo}/J* (mTmG; Jackson Labs, Bar Harbor, ME, Cat# 007676). In the resulting mouse, all insulin-producing cells (ventro-medial hypothalamus, pancreatic β- and δ-cells, data not shown), express fluorescent EGFP, while all non-insulin-producing cells express fluorescent tdTomato. We used only F1 generation mice for β-cell formation experiments. Stability of the mTmG mouse was gauged by culturing mTmG slices from both male and female mice. In addition, we also utilized the *B6. Cg-Tg(Ins1-EGFP)1Hara/J* (Ins1-GFP; Jackson Labs, Bar Harbor, ME, Cat# 006864) mouse for confirming and tracking β-cell generation in pancreatic slices. All mice were housed in specific pathogen-free (SPF) conditions at the DVR's animal care facility. For all experiments, mice were acclimated for 7–10 days prior to any experimental intervention. They were maintained on a 12 h light/dark cycle with ad libitum access to standard irradiated chow and filtered drinking water.

**Pancreatic slicing and culture conditions**. Tissue slicing: human pancreatic tissue biopsies were obtained from the cGMP facility at the Diabetes Research Institute (DRI), University of Miami, or as tissue/slices as part of the University of Florida and nPOD's human pancreatic tissue slice optimization initiative, the University of Alberta (Canada) IsletCore and Prodo Labs (Aliso Viejo, CA), all from appropriately consented donors. A complete de-identified demographics table is provided in Supplementary table 1. Mouse and human pancreatic slices were generated and processed as in[1]. Once cut, slices were washed with slice washing medium, which contains DPBS (Sigma Aldrich, St. Louis, MO, Cat# D8537) supplemented with 1X B27-minus Insulin (Invitrogen, Carlsbad, CA, Cat# A1895602), 1X penicillin-streptomycin-amphotericin B solution (Sigma Aldrich, St. Louis, MO, Cat# A5955), 100 µg/ml trypsin inhibitor from *Glycine max* (Sigma Aldrich, St. Louis, MO, Cat# T6522), 10 µg/ml aprotinin (Sigma Aldrich, St. Louis, MO, Cat# A6106), 10 µg/ml chymostatin (solubilized initially in DMSO; Sigma Aldrich, St. Louis, MO, Cat# 11004638001), and 5.5 mM D-glucose (Sigma Aldrich, St. Louis, MO, Cat# G8644). The addition of aprotinin and chymostatin was key to further prevent the degradation and maintain the viability and function of slices long-term, compared to earlier medium formulations for short-term studies[1]. Aprotinin inhibits serine proteases including trypsin, chymotrypsin, plasmin and kallikrein. Chymostatin is a strong inhibitor of chymotrypsin, papain, chymotrypsin-like serine proteinases, chymases and lysosomal cysteine proteinases such as cathepsins A, B, C, B, H, and L.

Dish coating: washed slices were placed atop 35 mm AirHive dishes (Biorep, Miami, FL) with the help of curved blunt forceps. Each AirHive dish was pre-coated with 250 µl collagen-fibronectin gel mixture. To prepare 1.6 ml of collagen-fibronectin solution (enough for six AirHive dishes at 250 µl/dish), 2.4 mg

fibronectin (Sigma Aldrich, St. Louis, MO, Cat# F1141) was added to 720 ml of DPBS for a 1.5 mg/ml final solution (it is suggested to make a master stock of the protein solution at 3.3 mg/ml for easy use; Sigma Aldrich, St. Louis, MO, Cat# D8537). 80 µl of 1 N NaOH (0.025 N final concentration; Sigma Aldrich, St. Louis, MO, Cat# S2770) and 800 µl of ~3 mg/ml collagen type-I (1.5 mg/ml final solution; Sigma Aldrich, St. Louis, MO, Cat# A10483-01) were added to the fibronectin solution. Dishes were coated with 250 µl/unit, making sure that the entire surface was covered by gently rotating the dish from side to side, and placement in a humidified cell culture incubator at 37 °C for 45–90 min prior to usage. Once coated, the dishes were carefully washed with ice-cold DPBS (Sigma Aldrich, St. Louis, MO, Cat# D8537) 3X times.

Medium and culture conditions for murine slices: washed murine slices were cultured using murine slice culture medium atop pre-coated AirHive dishes or Corning Transwell® polyester membrane cell culture inserts (Corning, NY; Cat #CLS3450-24EA) as controls. The formulation includes custom-made Waymouth's MB 752/1 medium (with L-glutamine; without D-glucose; Biological Industries, Cromwell, CT, Cat# 06-1110-01-1 A) containing 11 mM D-Glucose (Sigma Aldrich, St. Louis, MO, Cat# G8644), 1% heat-inactivated FBS (Invitrogen, Carlsbad, CA, Cat# 16140063), 100 µg/ml trypsin inhibitor from *Glycine max* (Sigma Aldrich, St. Louis, MO, Cat# T6522), 10 µg/ml aprotinin (Sigma Aldrich, St. Louis, MO, Cat# A6106), 10 µg/ml chymostatin (solubilized initially in DMSO; Sigma Aldrich, St. Louis, MO, Cat# 11004638001), and 1X penicillin-streptomycin-amphotericin B solution (Sigma Aldrich, St. Louis, MO, Cat# A5955). Dishes containing slices (maximum 3) and medium (maximum 1 ml) were placed in a humidified incubator at 37 °C for a maximum of 2 weeks. Medium changes were made every 8–12 h.

Medium and culture conditions for human slices: HPSs were cultured using human slice culture medium atop pre-coated AirHive dishes (coated similarly to those for murine slices) or Corning Transwell® polyester membrane cell culture inserts (Corning, NY; Cat #CLS3450-24EA) as controls. Human slice culture medium contains basal BrainPhys neuronal medium (Stemcell Technologies, Vancouver, BC, Cat# 05790) containing 2% B27 supplement minus-insulin (Invitrogen, Carlsbad, CA, Cat# A1895601), 1% penicillin-Streptomycin-Amphotericin B solution (Sigma Aldrich, St. Louis, MO, Cat# A5955), 1% Glutamax supplement (Invitrogen, Carlsbad, CA, Cat# 35050061), L-Glutamic Acid 3.7 µg/ml (Sigma Aldrich, St. Louis, MO, Cat# 49449), 5.5 mM final D-glucose concentration (of note, BrainPhys already contains 2.5 mM D-Glucose; Sigma Aldrich, St. Louis, MO, Cat# G8644), 100 µg/ml trypsin inhibitor from *Glycine max* (Sigma Aldrich, St. Louis, MO, Cat# T6522), 10 µg/ml aprotinin (Sigma Aldrich, St. Louis, MO, Cat# A6106), 10 µg/ml chymostatin (solubilized initially in DMSO; Sigma Aldrich, St. Louis, MO, Cat# 11004638001) and 1% HEPES buffer (Invitrogen, Carlsbad, CA, Cat# 15630080). Dishes containing slices (maximum 3) and medium (maximum 1 ml) were placed in a humidified incubator at 30 °C for a maximum of 2 weeks. Medium was changed every 8–12 h.

**Adenoviral vectors**. Recombinant adenoviruses were constructed using the serotype 5 adenovirus with an E1/E3 deletion. A codon optimized Cre variant was used. The Adv-RIP-Cre was created as in[25]. The SHIP was a gift from Dr. C. Brunicardi (U. of Toledo, OH). The SHIP construct was cloned upstream of an EGFP sequence to create the Adv-SHIP-EGFP adenovirus, and upstream of a codon optimized Cre to create an Adv-SHIP-Cre adenovirus. The reporter construct was a gift from Dr. P. Ravassard, Hôpital Pitié-Salpétrière-Paris, France, and was loaded onto an adenoviral packaging plasmid to create the Adv-(CMV)-LoxP-dsRED-STOP-loxP-EGFP adenoviral reporter. All adenoviral construction was performed at Vector Biolabs, Malvern, PA. Slices were transduced with a MOI of 50 for the reporter and a MOI of 10 for SHIP-containing viruses.

**Viability assessment**. Pancreatic slices were incubated with calcein-AM to indicate intracellular calcium activity (live cells) and ethidium homodimer-1 (dead cells). Addition of both of these reagents was done according to the manufacturer's recommendations as part of the Live/Dead viability/cytotoxicity kit for mammalian cells (Invitrogen, Carlsbad, CA, Cat# L3224). Once stained, slices were washed 3X times with DPBS (Sigma Aldrich, St. Louis, MO, Cat# D8537) and fixed with 3.7% paraformaldehyde. Fixed slices were counterstained with 1:400 4′, 6-diamidino-2-phenylindole (DAPI) (Thermo Fisher/Life Technologies, Waltham, MA, Cat# D1306) in DPBS (Sigma Aldrich, St. Louis, MO, Cat# D8537).

**Dynamic glucose-stimulated insulin/glucagon release studies**. Dynamic glucose-stimulated insulin/glucagon release (perifusion) assays were performed using pancreatic slices (3–4 slices per perifusion chamber in duplicates). Slices were perifused within a perifusion chamber (Warner Instruments, Holliston, MA, Cat# 64-0223) put together using silicone grease (Warner Instruments, Holliston, MA, Cat# 64-0378). The chamber rests on top of a chamber platform (Warner Instruments, Holliston, MA, Cat# 64-0281). Perifusion was done in Krebs buffer containing 125 mM NaCl, 5.9 mM KCl, 1.28 mM CaCl$_2$, 1.2 mM MgCl$_2$, 25 mM HEPES, and 0.1% bovine serum albumin, pH 7.4 at 37 °C, using a PERI3 machine with automated tray handling (Biorep Technologies, Miami, FL). Prior to loading slices into the chamber, slices were incubated in Krebs buffer containing 3 mM Glucose on a shaker set at 120 rpm at 37 °C, for 60–90 min. Slices were then loaded

in the chamber using a fine brush of 2/0 size (Electron Microscopy Sciences, Hatfield, PA, Cat# 66100-20) and connected to the perifusion system. Samples were challenged with either low or high glucose (G1 = 1 mM; G3 = 3 mM; G16.7 = 16.7 mM) or potassium chloride (KCl = 30 mM) at a flow rate of 100 μL/min. After 90 min of flushing in G3 solution, slices were stimulated with the following sequence: 16 min G3, 20 min G16.7, 30 min G1, 5 min G3 + KCl, and 15 min G3. Samples were collected every 60 seconds into a 96-well plate kept at 4 °C, while the perifusion chamber and perifusion solutions were maintained at 37 °C. After the perifusion, slices of one chamber were fixed in 4% paraformaldehyde for 30 min at room temperature and stored in PBS at 4 °C for insulin staining and subsequent β-cell quantification. Slices from the second chamber were placed in 500 μl of acid ethanol and stored at −20 °C. Perfusates were aliquoted and stored at −20 °C. Insulin and glucagon levels were determined using commercially available ELISA kits (Mercodia Inc., Winston Salem, NC, Cat# 10-1113-01 for insulin and Crystal Chem, Elk Grove Village, IL, Cat# 81520 for glucagon) as per manufacturer's recommendations with some modifications. Perfusates above the standard curve of the assay were diluted (2×–4X times depending on the secretion levels of the slice) to allow for quantification within the sensitivity range.

**Dynamic glucose-stimulated alpha-Amylase release studies**. Pancreatic amylase secretion was evaluated using 3X human pancreatic slices placed in a perifusion chamber. Perifusion was similar to that described for glucose secretion, with some modifications. Krebs buffer was supplemented with 10 μg/ml aprotinin (Sigma Aldrich, St. Louis, MO, Cat# A6106), 3 mM D-glucose (Sigma Aldrich, St. Louis, MO, Cat# G8644) and 1X penicillin-streptomycin-amphotericin B solution. Slices were challenged with either 3 mM Glucose + 10 pM Cholecystokinin 8 (Sigma Aldrich, St. Louis, MO, Cat# C2901), or 3 mM Glucose + 10 μM Carbachol (Sigma Aldrich, St. Louis, MO, Cat# 212385-M). Perifusates were collected after every 120 s in a 96-well plate kept on ice at all times. Amylase quantity was calculated using a calorimetric assay for amylase activity, based on the manufacturer's recommendations (BioVision, Milpitas, CA, Cat# K711-100).

**Quantification of cytosolic Ca$^{2+}$ levels**. Confocal images (pinhole = airy 1) of randomly selected islets were acquired on a Leica SP5 confocal laser-scanning microscope with 40× magnification (NA = 0.8). Pancreatic tissue slices were reconstructed in Z-stacks of 15–30 confocal images (step size = 2.5–5.0 μm) and analyzed using ImageJ. [Ca$^{2+}$]$_i$ responses were quantified as the areas under the curve of individual traces of Flou-4 fluorescence intensity during the application of stimuli. Pancreatic tissue slices were bathed in Fluo-4 AM (2 μM) for 1 h. To quantify changes in intracellular Ca$^{2+}$ levels, we selected regions of interest around individual endocrine cells and acinar cells. Fluorescence intensity was measured using ImageJ. Changes in fluorescence intensity are expressed as percentage changes over baseline ($\Delta F/F$). We measured changes in total cytosolic Ca$^{2+}$ levels by computing the area under the curve above baseline using Prism software (Prism 7, GraphPad software, La Jolla, CA). Areas under the curve were determined before, during, and after each stimulus for the same time period and compared with statistical tests. For quantification of [Ca$^{2+}$]$_i$ responses, we calculated the areas under the curve of the fluorescence intensity traces of Fluo-4. Our criteria for accepting [Ca$^{2+}$]$_i$ responses for analyses were that: (1) responses could be elicited ≥2× by the same stimulus; and (2) the peak signal was ≥2 times the baseline fluctuation. For quantification of β-cell responses, we selected cells that responded to increases in [Ca$^{2+}$]$_i$ during KCl and high glucose (16 mM) stimulation, in order to exclude endocrine α-cells from our quantification.

**Immunofluorescence analysis**. Tissue slices were washed 2× for 5 min in 1X phosphate buffered saline (PBS), pH 7.4 (Sigma Aldrich, St. Louis, MO, USA Cat# P3813), and fixed in 3.7% paraformaldehyde solution overnight. Slices were then washed 10X for 5 minutes in 1X phosphate buffered saline (PBS), pH 7.4 (Sigma Aldrich, St. Louis, MO, USA Cat# P3813). Permeabilization was done for 30 minutes using 0.3% Triton (Sigma Aldrich, St. Louis, MO, USA Cat# T9284-500ml) in 1X PBS, pH 7.4 (Sigma Aldrich, St. Louis, MO, USA Cat# P3813). After this, slices were blocked with blocking buffer containing [dH$_2$O, 5% normal donkey serum (Jackson Labs, Bar Harbor, ME, Cat# H-400), 0.1% Triton and 1X power block (Biogenex, San Ramon, CA, Cat# HK0855K)]. After 1 h, an additional blocking step was done for 10 minutes with serum-free protein block (Dako, now Agilent, Santa Clara, CA, Cat# X0909). Primary antibodies (see Supplementary Table 2 for list of antibodies) were dissolved in the blocking buffer and incubated for 48 h at 4 °C. Primary antibodies were then removed, and sections washed 7X for 5 min with 1X PBS, pH 7.4/0.1% Triton. Secondary antibody solutions (1:400) were made with Alexa Fluor 488, 594 or 647 donkey anti-primary antibody species and 1:400 4′, 6-diamidino-2-phenylindole (DAPI) (Thermo Fisher/Life Technologies, Waltham, MA, Cat# D1306) in blocking buffer. Samples were incubated with secondary antibodies for 90 minutes and then washed 7× for 5 min with 1× PBS, pH 7.4. To get the least amount of background fluorescence, an additional wash was performed overnight at 4 °C with 1X PBS, pH 7.4. For imaging, slices were transferred to 35 mm dishes containing a number-zero confocal grade crystal (MatTek Corporation, Ashland, MA, Cat# P35G-0-10-C). Care was taken to ensure slices were wet with DPBS/PBS at all times during confocal imaging (~100–150 μl on top of the slices to prevent drying). Imaging during culture was performed using

an ApoTome Axiovert 200M (Zeiss) fluorescent microscope, with the slices positioned in similar orientations as the preceding day while atop the AirHive. End-point imaging was performed using a Leica MP-NDD4/SP5/FCS/FLIM multiphoton/confocal upright F-techniques microscope MP/SP5 having a laser set at 30% intensity, pinhole set at 1, airy set at 1, with 10X and 20X magnification. Z-stack depth ranged between 120 and 130 μm depending on the slice thickness and condition, while the number of images in the Z-stack ranged between 15 and 25 with a step size ranging between 2.5 and 4 μm. To prevent bias, for eGFP/tdTomato and viability studies we utilized ImageJ/FIJI to perform quantitative analysis of cell populations, using total area calculations for fluorescent area and correlating to brightfield slice area. In additional quantitation of images, DAPI-positive were counted to determine the total number of cells, against which all the other markers were quantified.

**HIF1α determination**. ELISA (MyBioSource #MBS2885065) was performed to assess the levels of HNF1α comparing slices cultured in transwells and PFC. The results were normalized to total tissue protein content using a Pierce$^{TM}$ BCA assay (Thermo Scientific #23227).

**Quantitative real-time RT-PCR**. qRT-PCR[44] was performed with the Applied Biosystems/Thermo Fisher Scientific (Waltham, MA) TaqMan® assays listed in the Supplementary Data 1 (table of key resources).

**Statistical analysis**. Statistical analyses and graphing were performed using GraphPad Prism v8 (GraphPad software, La Jolla, CA). Following the Shapiro–Wilk normality test, statistical differences between groups were calculated by two-tailed paired t test or Wilcoxon signed-rank test, with 95% confidence intervals (∗$p < 0.05$; ∗∗$p < 0.01$; ∗∗∗$p < 0.001$). Results are expressed as mean ± SD. Statistical comparisons for cytosolic Ca$^{2+}$ levels were performed using Student's t test or one-way ANOVA followed by multiple-comparison procedures with the Tukey or Dunnett's tests. Data are shown as mean ± SEM.

**Key resources**. A table of key resources used in this manuscript is provided as Supplementary Data 1.

**Reporting summary**. Further information on research design is available in the Nature Research Reporting Summary linked to this article.

## Data availability
All relevant data are available from the authors. Source data underlying Figs. 1e–g; 2d–g, k; 3d–l, 4g–h, j–k; and 5b–d, g–h are provided as a Source data file.

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

## Acknowledgements

We acknowledge the contributions of Paul Latta and Michael Bellio (Ophysio, Inc.), Austin Frye, Maria Boulina, Elina Linetsky, Joel Szust, Armando Méndez, and the staff of the Imaging Core, cGMP facility, Preclinical Cell Processing and Translational Models Core, and the Clinical Chemistry, Biomarkers and Immunoassay Laboratory (all at the Diabetes Research Institute), as well as Ramón Poo and Felipe Echeverri (Biorep), who contributed to the design of Fig. 1a. Similarly, we extend our gratitude to Drs. C. Brunicardi and S-H. Liu (U. of Toledo) for the SHIP construct and advice. This research was performed with the support of the Network for Pancreatic Organ Donors with Diabetes (nPOD; RRID:SCR_014641), a collaborative type 1 diabetes research project sponsored by JDRF (nPOD: 5-SRA-2018-557-Q-R) and The Leona M. & Harry B. Helmsley Charitable Trust (Grant#2018PG-T1D053). This work was also supported by The Helmsley Charitable Trust George S. Eisenbarth nPOD Award for Team Science (2015-PG-T1D-052). The content and views expressed are the responsibility of the authors and do not necessarily reflect the official view of nPOD. Organ Procurement Organizations (OPO) partnering with nPOD to provide research resources are listed at http://www.jdrfnpod.org//for-partners/npod-partners/. Finally, we thank the donors and their families for their invaluable contribution to science. This work was funded by the Diabetes Research Institute Foundation (DRIF), the Inserra family, the Fred and Mabel R. Parks Foundation, the Tonkinson Foundation, ADA grant #1-19-ICTS-078 and NIH grants 1R43DK105655-01, 2R44 DK105655-02 and U01DK120393. In the context of the latter, we are especially thankful to our colleagues at the Human Islet Research Network (HIRN) for their collegial input and discussion. MMFQ was funded by an International Fulbright pre-doctoral fellowship/grant administered by the Foreign Fulbright Scholarship Board and the International Institute of Education. Drs. Domínguez-Bendala and Ricardo Pastori are the guarantors of this work and, as such, had full access to all the data in the study and take responsibility for the integrity of the data and the accuracy of the data analysis.

## Author contributions

M.M.F.Q.: discussion/advice on experimental design, data collection/analysis/interpretation and manuscript writing and preparation. J.W.: discussion/advice on experimental design, data collection/analysis/interpretation and manuscript writing and preparation. S.C.: discussion/advice on experimental design and data collection/analysis/interpretation. S.A-C.: discussion/advice on experimental design, and data collection/analysis/interpretation. A.T.: data collection/analysis/interpretation. J.A., D.K., and Y.M-H.: discussion/advice on experimental design and data collection/analysis/interpretation. H.H., M.B. and I.K.: generated human pancreatic slices. J.K.P.: discussion/advice on experimental design and data collection/analysis/interpretation. M.A.: discussion/advice on experimental design and co-leadership of nPOD's slice optimization initiative. S.S.: discussion/advice on experimental design and co-leadership of nPOD's slice optimization initiative. C.R.: discussion/advice on experimental design. A.P.: discussion/advice on experimental design and co-leadership of nPOD's slice optimization initiative. A.C.: discussion/advice on experimental design. C.F.: Discussion/advice on experimental

design, and data collection/analysis/interpretation. R.L.P.: conception/design of the study, data analysis/interpretation and paper writing. J.D-B.: conception/design of the study, data analysis/interpretation and paper writing.

## Competing interests

The University of Miami and Drs. Dominguez-Bendala, Fraker and Ricordi hold, but do not receive royalties for intellectual property used in this study. They are also equity owners in Ophysio, Inc., licensee of the intellectual property. The remaining authors declare no conflicts of interest.
