## [Peer Review File · Nature Communications]

Reviewers' Comments:

Reviewer #1:

Remarks to the Author:

This paper is a major breakthrough in pancreatic tissue long term culture of up to a remarkable 10 days, which is of sufficient duration for numerous applications in pancreatic islet and exocrine biology, but here it was applied to understanding neogenesis of beta cells from ductal cells using viral mediated lineage tracing in normal pancreas, and to lesser extent also in T1D and T2D human pancreas. I strongly believe that this technical capability is extremely important and useful to the large community of pancreatic investigators and should be disseminated. I also think that the data on BMP induced neogenesis is novel. I however would like some improvements and more rigor in some of their data.

Figure 1. The data shown here are very convincing that the pancreatic slices are much more viable particularly after Day 5 with the PFC wells than the control trans wells. Along with these functional studies, an oxygen-deficient environment could induce increased expression of hypoxia-inducible factors which would also be alleviated in PFC wells.

Figure 2. 2j. The purported evidence of islet health at D10 is that the respective cell type immunolabeling remains discrete in the PFC and much less (or as they said - lower defined boundaries) so with the transwell. Is this because of cell death or anoxic injury, some de-differentiation. EM on D10 with characteristic features of each cell (granule appearance) and organelle injury would have made these conclusions more apparent.

2m: It is difficult to assess from these images the acinar intactness and injury. High power views to show more obvious acinar morphology with amylase (in zymogen granules) concentrated in the apical poles of cells within an acinus should be shown. Acinar injury is typically shown as intracellular vacuole formation (from perturbed autophagy), large blebs forming on the basal membrane, junctional complex disruption, and necrotic markers.

Figure 3. 3a: in this pancreatic slice perfusion study, the insulin content of the slice should have been determined, and customarily the secreted insulin should be expressed as percentage of total tissue insulin content.

3d: Why in control acute 1 mM glucose stimulation secreted more insulin than 16mM?

3f: Error bars missing

3h: No legend as to which trace correspond to what. Was this single beta cell or whole islet response? Were the beta cells within the whole islet at Day 10 synchronous and oscillatory in calcium response like Day 0 or was there disruption in coupling. How about the TW, although little response to 18G were the calcium response just weaker, in fewer cells within the islet, or loss of synchrony.

3k: Similar comments as 3h. Calcium responses in acini are observed as sequential light up of all acinar cells and oscillatory in pattern- was this pattern observed at Day 10. This is important for both islet and acini as this demonstrates that the coupling of beta and acinar cells remain intact at Day 10

3i: no labels for each trace. Author should put the acute sample for control as other figures.

Figure 4 (mouse) and Figure 5 (human) are very impressive in the longitudinal tracking of beta cell neogenesis induced by BMP-7 especially after alloxan destruction of the mouse beta cells. The techniques employed were likewise clever and impressive. Fig. 4c: They showed that insulin+ cells were generated de novo from ductal cells, but did not use a ductal cell marker, and should do so to be truly convincing that these beta cells genuinely differentiated from the ductal cells.

BMP7 induction of beta cell neogenesis in the T1D and T2D human islets in the supplementary figures should be shown as a main figure.

Methods. A bit description of the differences in the buffer that protected the pancreatic slices.

Good that they did mention using the better protease inhibitors - aprotinin and chymotstatin.

Reviewer #2:

Remarks to the Author:

This manuscript reports an improvement in culturing pancreatic slices in their native-like environment for long-term applications. First, the author designed a novel culture system that provides an improved system to deliver oxygen to the pancreatic explants. Then they characterized cell viability and function of endocrine and exocrine pancreas in this culture system during a 10-day time-period. Finally, they provided data on the potential application of this improved system to study beta cell regeneration in mouse and human. The improvement in the maintenance of the culture system for longer-time is of great importance to study physiological and pathological processes of human pancreas. This definitely highlights the importance of this study. However, the study lack of substantial experimental setting to support the scientific claims. In particular, the evidence indicating beta cell neogenesis using this model is not sufficient. Therefore, several key experiments and further evidence are required to clarify and support the finding of this manuscript.

Major points

-Using a modeling prediction, the author estimated and characterized the required oxygen for each type of culture systems. However, there is no direct experimental data indicating that oxygen is the only parameter that improved the maintenance of the culture. I was wondering if the authors can support this claim experimentally, for instance, through culturing PFC system in different levels of oxygen and analyze metabolic and morphological features.

-The characterization of the new system has been done mainly on immunohistochemistry (IHC). To provide further supporting evidence, the authors need to check some key markers (such as transcription factors) for different types of endocrine and exocrine cells. This can be done using IHC, qPCR analysis or western blotting. This analysis are important to better characterize the changes between TW and PFC systems that results in improvement of PFC culture. For instance, in Fig2c, in TW condition more dead cells are shown, however, it remains unclear, which cell type are undergo cell death, as total endocrine cells (Fig2K) and exocrine cells look normal between two systems. Therefore, it is important to check some apoptotic markers contained with different types of pancreatic cells. Additionally, no data is shown on the maturity of beta cells in tow systems. Only showing Insulin or Glucagon does not clearly indicate the presence of mature functional beta or alpha cells in the system. Finally, in supporting loss of islet integrity and its improvement in PFC vs TW, more analysis needs to be presented. For instance, high-resolution imaging of E-cadherin and/or Beta-catenin might indicate cell-cell interaction changes between two systems.

-On the functional analysis of HPSs, regarding the data presented in Fig3, the freshly sectioned (acute) samples show lower GSIS. Of course the authors discuss that this might be due to stress of sectioning and so on, but I am really wondered why they didn't rest the samples for a day and then perform the experiments. The problem here is the HPSs cultured 10 days in TW perform similar GSIS to the control acute samples. For having proper control at least they should compare TW and PFC samples at day 1 vs day 10.

-In Figure 3a, in the PFC culture the total secreted insulin is higher even in the KCl. Is this indicating that the total insulin levels is higher in this condition, or we have more insulin granules? This can be very important as in later figures the authors show increase in insulin+ cells in the PFC condition. They should clearly rule out if PFC condition increase total levels of insulin or insulin granules that might contribute to increased EGFP signals in later figures.

-In figure 4 the authors claim that PFC culture of mouse slices result in islet neogenesis upon treatment with BMP-7 during 10 day culturing. They used slices from InsCre, mTmG mice and

showed increased yellow cell population as well as EGFP signal upon treatment with BMP-7 indicating conversion of non-beta to beta cells. One important point here is, there is no evidence that the observed process is neogenesis. The term of neogenesis refers to regeneration of beta cells from progenitors. There is not supporting data to show the existence of progenitors and their differentiation towards beta cells. To claim this, the authors need at least to show the presence of Neurog3+ cells in their condition. Therefore, what is presented is most likely the conversion of non-beta to beta cells that can be referred to general term of regeneration or cell conversion but not neogenesis. Furthermore, no high resolution data is shown to indicate that the yellow cells are derived from the ductal epithelium. For instance, in FigureC, lower part a yellow cell population is shown that is far away from the ductal region. If such cells are derived from duct is it possible that they move that far in a short period within the slice culture? For me the presented data are of low quality and to show that there is cell conversion in this system the authors need to provide high resolution picture with high magnification involving different pancreatic markers. This might provide more information about the origin of the converted cells. If they cannot provide any data regarding the presence of Neurog3+ cells they need to revisit using the neogenesis term.

-The same concept also applies to the figure 5 on human slices. I do understand the limitations with lineage tracing in human systems and I appreciate the design of this system in figure 5. However, the conclusion claimed by the authors here might not truly reflect the presented data. Again, there is no data showing the presence of NEUROG3+ progenitors. The low quality and magnification of the presented data makes it very difficult to monitor the actual process of cell conversion in this system. Most importantly, the authors failed to show that treatment of BMP7 result in increase in yellow population statistically. For me increased levels of EGFP is not enough to claim increase in total beta cells. First, the data presented in fig5 c and h showing the % of slice expressing EGFP but do not show if there is increase in beta cell number. Second, there is no data to rule out that treatment of BMP7 does not increase insulin expression and levels.

Other points

-introduction: "...are more likely to reflect the native biology of the organ" not completely true. They are the closest system to the in vivo condition so far, but till which degree they reflect native biology of the organ is not clear.

-Fig1b-d. Could this analysis be extended to 10 days? This would help to monitor the system better over 10 day experimental course presented in later figures.

-Page 7: "... in transwells have less-defined boundaries (Fig. 2j) and with less defined boundaries." Repeated wording.

-Some figures miss scale bar.

-Fig2K. Are all these cell alive? Is there any analysis that can show the live vs dead cells in endocrine compartments?

-Fig3g,I, What is 3G, please indicate it in the legend.

-Fig3i, where is the reference or control sample at day 1?

-Fig3K-m, here only shows that the experiment technically work. I found the actual data regarding the PFC and TW cultures in the supplementary data. I would suggest to put those data in the main figure.

-The experiment on mouse slices is done using pancreas from 6-8-week-old mice. This is relatively young age for mice. Therefore, the pancreatic lineage might still retain higher degree of plasticity. I am curious if old animal also present the formation of new beta cells when treated with BMP-7 in the slice culture. This would be more relevant for diabetes condition in older population.

-Page13: "... due to the treatment of slices with BMP7." They mean "alloxan"?

-Fig4c, is the lower picture cropped from the upper part? For me it doesn't seem to be.

-Fig4i. The green signal is barely detectable. The same for Fig4f.

-Page16: "We could also detect several regions of non- β -cells transitioning to EGFP-expressing cells, mirroring previous observations using mouse slices (Fig. 5a)." I don't see this transitioning in the picture. High magnification and resolution is required.

-Fig5d, is not mentioned within the text.

-The conclusion section especially on neogenesis is too strong. It needs to be written more based on the actual data.

-Figure S5, the control and MBP-7-treated samples present similar levels of yellow cells. While the EGFP signal is different between the two condition. This might indicate the possible effect of BMP-7 on insulin levels rather than increase in cell conversion. Similar data are also presented in Fig5 c and h.

Reviewer #3:

Remarks to the Author:

In this manuscript Qadir et al set out to develop novel methodology for long term culture and subsequent examination of function and islet cell regeneration using human pancreatic slices. Authors report successful development of enhanced pancreatic slice culture system that allows for improved oxygenation, slice viability and endocrine functionality (i.e. glucose and KCl-stimulated insulin secretion). Furthermore, investigators utilized adenoviral vectors to perform longitudinal lineage tracing tracking of potential beta cell neogenesis in response to the treatment with BMP7, a compound previously shown by this group to induce human beta cell neogenesis from exocrine cells in the pancreas (i.e. purported ductal cell progenitors).

Critique:

1. Figure 1: Authors state that parameters used to model oxygen consumption are estimated based on values reported in references 15. The data in reference 15 however reports data from isolated/cultured porcine acinar cells and islets kept in culture from 0 to 8 days. It's not clear why this data set was chosen to approximate/model oxygen consumption in intact human slices given clear differences in tissue handling and also reported differences between porcine and human islets/exocrine tissue. In addition, lactate production (panel e) data is presented (in hours) which is inconsistent with data presentation in other graphs (i.e. in days). Furthermore, authors need to include units for Fig 1e and 1f (i.e. lactate production and glucose consumption). Finally, the claim that 'enhanced oxygenation' preserves oxidative phosphorylation is not supported by presented data. Decreased "glucose consumption" in PFC group can be attributed to numerous factors and cellular changes and for example may reflect attenuated metabolic rate or changes in cellular proliferation/survival. Authors need to perform additional experiments to directly demonstrate enhanced mitochondrial function/oxygen consumption/oxidative metabolism or use glucose carbon isotope techniques to demonstrate enhanced glucose oxidation under PFC condition.

2. Figure 2. Given focus of the manuscript on endocrine pancreas biology, authors should perform additional studies of endocrine pancreas viability/expression in PFC vs. TW culture conditions. This should include examination of cell turnover (proliferation and apoptosis) in beta and alpha cells. This will be particularly important given the focus on beta cell regeneration and turnover. In addition, author should examine expression of key maturation markers for beta (e.g. MAFA, NKX6.1, PDX-1 etc) and alpha (e.g. ARX) cells. This is important given dramatic increases in insulin secretion reported in Fig. 3 under PFC conditions. Finally, since authors suggest enhanced oxygen delivery and improved mitochondrial oxidation in PFC group, staining for hypoxia-regulated transcription factors (e.g. HIF1, c-JUN etc) and/or markers of mitochondrial integrity will provide supporting evidence for the stated hypothesis.

3. Figure 3. Authors report enhanced glucose-stimulated insulin secretion under perfusion conditions with 10 day culture using PFC dishes. Although this data is impressive, there are number of questions that arise from these studies. Firstly, contrary to what authors state, none of

the groups display "classic biphasic pattern" of glucose-stimulated insulin release. In PFC group, insulin concentrations continuously rise during 16mM glucose period without clear 1st/2nd phase distinctions. The physiological significance of pancreas slice perfusion will be reinforced by data demonstrating physiological release of other important endocrine hormones such as glucagon and somatostatin. Finally, PFC group has a significant increase in glucose-independent insulin release to KCl. KCl is commonly used as a measure of maximal insulin output and used as a marker of total beta cell mass/numbers. Authors should comment on the reason for increased KCl-stimulated insulin release in PFC group despite comparable levels of insulin positive cells reported in sections from PFC vs. TW slices.

4. Figures 4 and 5. Data on longitudinal tracking of purported beta cell regeneration in mouse/human slices is intriguing, but is largely preliminary and not supportive of main conclusions of the study. Firstly, to prove generation of "newly" formed beta cells (with BMP-7) authors need to show clear increase in overall beta cell numbers (or other measures of insulin + area). These data need to show that newly formed cells express key beta cell maturation transcription factors (e.g. NKX6.1, PDX1 etc) as well as other markers of functionality. Demonstrating that treatment with BMP-7 also positively alters insulin secretion in perfusion setting (especially KCl-mediated insulin release) will also significantly strengthen the argument that beta cell neogenesis has occurred in this model. In addition, the concept of beta cell regeneration implies replacement of lost beta cells, however current study in human slices examines response of otherwise "healthy" beta cells to a supra-physiological stimuli of BMP-7. Authors need to clearly acknowledge that there are limitations of this approach in human tissue and the study of human beta cell physiological neogenesis is not possible using this model. Indeed, repeated studies in mice and humans have demonstrated lack of any evidence for physiological occurrence of beta cell neogenesis.

REVIEWER 1

NOTE: corrections in the manuscript are highlighted in yellow.

This paper is a major breakthrough in pancreatic tissue long term culture of up to a remarkable 10 days, which is of sufficient duration for numerous applications in pancreatic islet and exocrine biology, but here it was applied to understanding neogenesis of beta cells from ductal cells using viral mediated lineage tracing in normal pancreas, and to lesser extent also in T1D and T2D human pancreas. I strongly believe that this technical capability is extremely important and useful to the large community of pancreatic investigators and should be disseminated. I also think that the data on BMP induced neogenesis is novel. I however would like some improvements and more rigor in some of their data.

We thank reviewer 1 for his/her overall praise of our findings and insightful comments. We agree that the technical capability herein described will be important and useful to the community of pancreatic investigators at large. In general, we appreciate the suggestion of additional studies that would undoubtedly enhance the depth of our manuscript. In this context, it is important to stress that this work was conducted within a consortium, the nPOD Slice Working Group, which originally involved 5 teams. Our manuscript lists the leaders of these (Speier, Atkinson, Pugliese, Caicedo), who have had an active input throughout. Organs for slicing were centrally processed either in our lab (Miami, FL) or the nPOD lab (Gainesville, FL). Over the past 3 years, only a limited number of slices/donor organ was made available to each one of these teams for specific experiments. Even the slices that we generated in our lab were distributed to the other consortium teams as well. Thus, for each donor, we had to carefully maximize the number of assays to be performed; and, when slices were insufficient, prioritize such assays. In other words, the scope of our research has been dictated to a large extent by the available number of slices and —needless to say— the irregularity of appropriate offers of human pancreata for research. To this, we have to add that not all samples were suitable for slicing, particularly if the pancreas was fatty (a very common occurrence) or fibrotic.

This is all to say that, while the reviewers propose some very worthy experiments to expand even further the breadth of our findings (many of which already in our pipeline), conducting all of them with an appropriately powered number of samples would take a very long and unpredictable amount of time. Our manuscript is intended as a first report of a new technology to study the human pancreas, not as a comprehensive opus that provides all the answers. We hope that, by making this initial report available to the scientific community, others will also be able to exploit our technology to the fullest extent.

Below we address the specific comments/suggestions:

- 1. Figure 2. 2j. The purported evidence of islet health at D10 is that the respective cell type immunolabeling remains discrete in the PFC and much less (or as they said - lower defined boundaries) so with the transwell. Is this because of cell death or anoxic injury, some de-differentiation. EM on D10 with characteristic features of each cell (granule appearance) and organelle injury would have made these conclusions more apparent.***

R: We agree that granular appearance of different cell subtypes may shed additional light on the processes going on in degrading slices. However, it is important to emphasize that we don't

claim the immunofluorescence analysis to be the main evidence of islet health at day 10. It is the actual functional studies reported on figure 3 that support that claim.

- 2. 2m: It is difficult to assess from these images the acinar intactness and injury. High power views to show more obvious acinar morphology with amylase (in zymogen granules) concentrated in the apical poles of cells within an acinus should be shown. Acinar injury is typically shown as intracellular vacuole formation (from perturbed autophagy), large blebs forming on the basal membrane, junctional complex disruption, and necrotic markers.**

R: We agree that the resolution of the immunofluorescence images presented in figure 2m is insufficient to ascertain exocrine cell injury/intactness, and in fact we state that “*No apparent differences between day 0 and day 10 on either culture condition could be observed in the IF patterns of exocrine markers*”. We have now added in our discussion of the data [*“Long term viability (...)*” Results section] the caveat that the resolution of the imaging is insufficient to detect potential ultrastructural changes such as the ones described by the reviewer. This initial manuscript, while also presenting some data on exocrine tissue functionality, is focused mainly on islets owing to the application we have chosen to describe (new β -cell formation). However, we are planning to expand these studies to the exocrine compartment in collaboration with nPOD partners with that particular expertise. Those studies, for which a significant number of new slice donors will be required, will be the subject of a future report.

- 3. Figure 3. 3a: in this pancreatic slice perfusion study, the insulin content of the slice should have been determined, and customarily the secreted insulin should be expressed as percentage of total tissue insulin content.**

R: This is a good suggestion, but for normalization purposes we presented instead the amount of DAPI⁺/INS⁺ cells/slice (showing that it was similar in all three groups). Unfortunately, we cannot go back to these slices for the suggested normalization. We have made this clear in the relevant results section.

- 4. 3d: Why in control acute 1mM glucose stimulation secreted more insulin than 16mM?**

R: Acute slices clearly did not show normal glucose responsiveness. This is also typical of islets if perfusion is conducted immediately after isolation, as indicated in the discussion. One of the lessons to be learned from these experiments is that HPSs should not be used in the immediate peri-slicing period, as also discussed.

- 5. 3f: Error bars missing**

R: There are no error bars in this graphic because this is one chamber’s perfusion. There are 3 slices in it but there is only one chamber. We could not use more chambers because this was a T1D sample, of which there is an exceedingly scarce supply that has to be shared among all the consortium members. We had only a very few slices and a minimum of 3 slices/perfusion chamber is needed. Therefore, these data are representative of one run. This has been clarified in the legend of the figure.

- 6. 3h: No legend as to which trace correspond to what. Was this single beta cell or whole islet response? Were the beta cells within the whole islet at Day 10 synchronous and oscillatory in calcium response like Day 0 or was there disruption in coupling. How about the TW, although little response to 18G were the calcium response just weaker, in fewer cells within the islet, or loss of synchrony?**

R: These 3 arrows correspond to the 3 endocrine cells which are being pointed to by arrows in figure 3g, and which are now color-coded to match the graphic in 3h. We thank the reviewer for pointing out this omission.

We found the β -cells to be synchronous and oscillatory. We did not observe a disruption in coupling. Please refer to the supplemental video for visualization of the pulses. The β -cells in TW were synchronous and oscillatory as well. Overall, pulses in cultured transwell-cultured slices during 16G stimulation were weaker, but the results were not significantly different. This is now indicated in the text [*Enhanced oxygenation is supportive* (...) Results section].

- 7. 3k: Similar comments as 3h. Calcium responses in acini are observed as sequential light up of all acinar cells and oscillatory in pattern- was this pattern observed at Day 10. This is important for both islet and acini as this demonstrates that the coupling of beta and acinar cells remain intact at Day 10.**
3i: no labels for each trace. Author should put the acute sample for control as other figures.

R: Arrows pointing at acinar cells are now color-coded to represent the traces in the graphics in 3l and 3m. We thank the reviewer for pointing out this omission. The traces are from single acinar responses shown in the panels of figure 3k. The acinar activity was synchronized during stimulation with carbachol. This is also now indicated in the text.

Regarding the acute control for figure 3i: These experiments were done in the context of an nPOD-sponsored consortium where our lab was charged with the conduct of long-term studies, leaving acute measurements to other partners. Slices from each donor were generated centrally and distributed to each lab for specific experiments. While we could use a few donors for acute perfusion experiments whenever slices were generated in Miami (fig. 3a), all acute Calcium imaging experiments were conducted in other labs. This is why acute is not shown in this graphic. At any rate, the entire second half of the figure basically aims at confirming that culture doesn't change significantly the expected physiology of glucose-mediated insulin secretion. This is now indicated in the manuscript [*Enhanced oxygenation is supportive* (...) Results section].

- 8. Figure 4 (mouse) and Figure 5 (human) are very impressive in the longitudinal tracking of beta cell neogenesis induced by BMP-7 especially after alloxan destruction of the mouse beta cells. The techniques employed were likewise clever and impressive. Fig. 4c: They showed that insulin+ cells were generated de novo from ductal cells, but did not use a ductal cell marker, and should do so to be truly convincing that these beta cells genuinely differentiated from the ductal cells.**

R: The reviewer makes an excellent point: indeed, we cannot conclusively establish in these experiments that the origin of these insulin-expressing cells is exclusively ductal. We acknowledge that the approach used to set up the technique is not designed to answer that question. To unequivocally determine that, additional lineage tracing experiments (e.g., using an ALK3- or CK19-driven Cre recombinase + reporter) would have to be done. An extensive battery of such experiments is in our pipeline, but we consider that those fall beyond of the scope of this initial communication, which is centered on the technical aspects of long-term slice culture and our ability to conduct adenoviral-mediated transduction for the monitoring of cell fate changes —with β -cell regeneration experiments presented only as an example of the many potential applications of this novel technology.

9. BMP7 induction of beta cell neogenesis in the T1D and T2D human islets in the supplementary figures should be shown as a main figure.

R: We have done this now (Figure 5).

10. Methods. A bit description of the differences in the buffer that protected the pancreatic slices. Good that they did mention using the better protease inhibitors - aprotinin and chymostatin.

R: Indeed, aprotinin and chymostatin, in addition to the already described soybean trypsin inhibitor, made a difference in preserving the integrity of the acinar tissue and prevent excessive degradation. Aprotinin inhibits serine proteases including trypsin, chymotrypsin, plasmin and kallikrein. Chymostatin is a strong inhibitor of chymotrypsin, papain, chymotrypsin-like serine proteinases, chymases and lysosomal cysteine proteinases such as cathepsins A, B, C, B, H, and L. This is now detailed in the Methods section.

REVIEWER 2

NOTE: corrections in the manuscript are highlighted in yellow.

We thank reviewer 2 for highlighting the importance of our study, and most requests are indeed on point and likely to improve the accuracy of our report. Below we address the specific comments/suggestions:

- 1. Using a modeling prediction, the author estimated and characterized the required oxygen for each type of culture systems. However, there is no direct experimental data indicating that oxygen is the only parameter that improved the maintenance of the culture. I was wondering if the authors can support this claim experimentally, for instance, through culturing PFC system in different levels of oxygen and analyze metabolic and morphological features.***

R: The reviewer makes a good point. Our mathematical modeling suggests that 21% oxygen approximates the best situation possible based on published OCR for endocrine and exocrine compartments, the tissue thickness and area, and other published parameters. However, we did not test multiple oxygen concentrations experimentally. This work was conducted as part of a consortium effort (nPOD Slice Working Group), in which a rather reduced number of slices from each donor were made available to several groups (including those generated in-house). The paucity of tissue from each donor, and the wide scope of experiments to be conducted for each, made it advisable to choose one setting alone (the one predicted to be the best) in order to maximize our resources. In previous work on islets (Fraker *et al*, 2013; PMID: 23068091), we did validate experimentally mathematical predictions for specific O₂ concentrations. Therefore, while we cannot categorically exclude that other oxygen concentrations may result in even better outcomes, we are confident in the robustness of our prediction.

- 2. The characterization of the new system has been done mainly on immunohistochemistry (IHC). To provide further supporting evidence, the authors need to check some key markers (such as transcription factors) for different types of endocrine and exocrine cells. This can be done using IHC, qPCR analysis or Western blotting. These analyses are important to better characterize the changes between TW and PFC systems that results in improvement of PFC culture.***

R: As indicated above, in the context of the nPOD Slice Working Group consortium, only a limited number of slices/donor organ was made available to each one of these teams for specific experiments over the past three years. Even the slices that we generated in our lab were distributed to the other consortium teams as well. Thus, for each donor, we had to carefully maximize the number of assays to be performed; and, when slices were insufficient, prioritize such assays. In other words, the scope of our research has been dictated to a large extent by the available number of slices and —needless to say— the irregularity of suitable offers of human pancreata for research. This is all to say that, while the reviewer proposes some very worthy experiments to expand even further the breadth of our findings (many of which already in our pipeline), conducting all of them with an appropriately powered number of samples would take a very long and unpredictable amount of time. We hope that, by making this initial report available to the scientific community,

others will also be able to exploit our technology to the fullest extent. Having said that, we have been able to secure one more human sample suitable for slicing, and we have used the limited resulting slices to run of some of the experiments suggested by the reviewers. At the suggestion of the reviewer, we provide now a qRT-PCR analysis for some key pancreatic markers conducted on this sample after 10d of culture in either condition. The results are expressed as relative quantification of expression in PFC vs. TW (where the latter is =1; $\Delta\Delta C_t$ method; sequential slices assigned to each group). All the pancreatic markers that we examined are upregulated in the PFC vs. the TW group, further confirming the conclusions from our IF and functional studies. This is now shown in **Fig. S1d**.

3. *For instance, in Fig2c, in TW condition more dead cells are shown, however, it remains unclear, which cell type undergo cell death, as total endocrine cells (Fig2K) and exocrine cells look normal between two systems. Therefore, it is important to check some apoptotic markers contained with different types of pancreatic cells.*

R: Fig. 2k presents the % of each endocrine type vs. all the cells in the slice. As shown there, no differences in the overall proportion of beta, alpha or delta cells can be observed between the different groups. However, the islet area is lower in TW at 10 days (Fig. 2I). It is therefore implicit that the whole slice must also have a smaller area (otherwise, the proportion of endocrine cells vs. all the cells in the slice would be smaller in day 10 TW). This is indeed the case, as viability (2d, e) and overall cross-sectional area are reduced at day 10 TW. In short, these data suggest that cell death occurs in all cell types in the same relative proportion. We have added a statement to this effect in the manuscript [“Long-term viability (...) Results section].

4. *Additionally, no data is shown on the maturity of beta cells in two systems. Only showing Insulin or Glucagon does not clearly indicate the presence of mature functional beta or alpha cells in the system. Finally, in supporting loss of islet integrity and its improvement in PFC vs TW, more analysis needs to be presented. For instance, high-resolution imaging of E-cadherin and/or Beta-catenin might indicate cell-cell interaction changes between two systems.*

R: We agree that IF alone cannot be used to predict maturity or function of specific islet cells. However, we present an array of functional assays in figure 3 showing that long-term PFC-cultured slices exhibit near-normal glucose-responsive insulin secretion, as opposed to those cultured in

standard transwells. The new qRT-PCR data also show upregulation of key endocrine functional markers, including MAFA, GLUT-1 (SLC2A1) and Chromogranin A (CHGA).

Unfortunately, not all pancreatic samples are suitable for slicing. Fatty/fibrotic pancreata present challenges for slicing, and very often the resulting slices are of insufficient quality for ultrastructural analysis. However, EpCAM and CADH1 (E-cadherin) are upregulated in the PFC vs. the TW group, as shown in the new qRT-PCR data. Future lines of research will focus on the optimizing ultrastructural analysis of human pancreatic slices.

- 5. On the functional analysis of HPSs, regarding the data presented in Fig. 3, the freshly sectioned (acute) samples show lower GSIS. Of course, the authors discuss that this might be due to stress of sectioning and so on, but I really wonder why they didn't rest the samples for a day and then perform the experiments. The problem here is the HPSs cultured 10 days in TW perform similar GSIS to the control acute samples. For having proper control at least, they should compare TW and PFC samples at day 1 vs day 10.***

R: The reviewer makes a valid point, and we acknowledge the possibility that resting the slices for one extra day may result in better acute responses. However, this was our experimental design when we first set out to conduct these studies, and we kept it that way for consistency. We must stress, though, that the main comparison to be studied in our experimental design is between day 10 TW and day 10 PFC. The purpose of our manuscript is to enable the long-term culture of human pancreatic slices for a variety of studies. Redoing the acute experiments with an additional day of rest would require no less than 3 additional donors (whose availability is completely unpredictable), and the information to be gathered would not change the main conclusions of our report. However, we have modified the discussion to reflect the point raised by the reviewer.

- 6. In Figure 3a, in the PFC culture the total secreted insulin is higher even in the KCl. Is this indicating that the total insulin level is higher in this condition, or we have more insulin granules? This can be very important as in later figures the authors show increase in insulin+ cells in the PFC condition. They should clearly rule out if PFC condition increase total levels if insulin or insulin granules that might contribute to increased EGFP signals in later figures.***

R: Indeed, since these data are normalized to the number of insulin-producing cells, it follows that PFC-cultured slices do respond better not just to glucose stimulation but also to KCl-induced membrane depolarization on a per-cell basis. We hypothesize that this is due to the impairment of β -cells cultured for 10 days in TW conditions. However, the reviewer states that this may influence the results presented in later figures. Figures 4 and 5 present experiments conducted only on PFC, and the increase in EGFP signal is genetically mediated, i.e., as a result of the activation of the insulin promoter. Such increase is measured when the slices are exposed to BMP-7 and compared to that observed in the absence of BMP-7 (control). Since both BMP-7 and controls are cultured in PFC membranes, the net increase in EGFP signal that we observe in the BMP-7 group is not related to PFC culture.

- 7. In figure 4 the authors claim that PFC culture of mouse slices result in islet neogenesis upon treatment with BMP-7 during 10-day culturing. They used slices from InsCre, mTmG mice and showed increased yellow cell population as well as EGFP signal upon***

treatment with BMP-7 indicating conversion of non-beta to beta cells. One important point here is, there is no evidence that the observed process is neogenesis. The term of neogenesis refers to regeneration of beta cells from progenitors. There is not supporting data to show the existence of progenitors and their differentiation towards beta cells. To claim this, the authors need at least to show the presence of Neurog3+ cells in their condition. Therefore, what is presented is most likely the conversion of non-beta to beta cells that can be referred to general term of regeneration or cell conversion but not neogenesis. Furthermore, no high-resolution data is shown to indicate that the yellow cells are derived from the ductal epithelium. For instance, in Figure C, lower part a yellow cell population is shown that is far away from the ductal region. If such cells are derived from duct is it possible that they move that far in a short period within the slice culture? For me the presented data are of low quality and to show that there is cell conversion in this system the authors need to provide high resolution picture with high magnification involving different pancreatic markers. This might provide more information about the origin of the converted cells. If they cannot provide any data regarding the presence of Neurog3+ cells they need to revisit using the neogenesis term. The same concept also applies to the figure 5 on human slices. I do understand the limitations with lineage tracing in human systems and I appreciate the design of this system in figure 5. However, the conclusion claimed by the authors here might not truly reflect the presented data. Again, there is no data showing the presence of NEUROG3+ progenitors. The low quality and magnification of the presented data makes it very difficult to monitor the actual process of cell conversion in this system.

R: The reviewer is correct. As also indicated to reviewer 1, we cannot conclusively establish in these experiments that the origin of these insulin-expressing cells is exclusively ductal, or progenitor-derived. This is a hypothesis based on our earlier body of work where we characterized BMP-7-responsive ductal-resident cells. However, we acknowledge that the approach used to set up the technique is not designed to answer that question. For that, additional lineage tracing experiments would have to be done. A comprehensive battery of such experiments is in our pipeline, but we consider that those fall beyond of the scope of this initial communication, which is centered on the technical aspects of long-term slice culture —with beta cell regeneration experiments presented only as an example of the many potential applications of this novel technology. At any rate, we have reworded the manuscript throughout for accuracy according to the suggestion of the reviewer. We have also eliminated the term neogenesis until there is future confirmation of such from additional lineage tracing experiments, and reflected this caveat in the Discussion.

- 8. Most importantly, the authors failed to show that treatment of BMP7 result in increase in yellow population statistically. For me increased levels of EGFP is not enough to claim increase in total beta cells. First, the data presented in fig5 c and h showing the % of slice expressing EGFP but do not show if there is increase in beta cell number. Second, there is no data to rule out that treatment of BMP7 does not increase insulin expression and levels.*

Figure S5, the control and MBP-7-treated samples present similar levels of yellow cells. While the EGFP signal is different between the two condition. This might indicate the

possible effect of BMP-7 on insulin levels rather than increase in cell conversion. Similar data are also presented in Fig5 c and h.

R: The increase in EGFP population is statistically significant in the BMP-7 group vs. control. The increase in the “yellow” population (i.e., those cells that are transiently both EGFP- and dsRED-positive, reflecting new onset of insulin expression) is not statistically significant between the two groups, but it is important to stress that, while green signal is cumulative, yellow is not. Unlike the green signal, the yellow that we detect at the time of imaging may or may not be representative of what happens over a 24h period. Therefore, determining “true” statistical significance based on the arbitrary measurement of cells that are only yellow for a short period of time may not be possible. This has now been added to the discussion.

Regarding the potential effect of BMP-7 on islets, the green signal is not lower or higher depending on the levels of expression of insulin. Whenever the insulin promoter is expressed in a cell containing both the reporter and the SHIP-Cre, the EGFP signal will be activated under the control of the CMV promoter (see fig. 4d). In other words, a β -cell expressing very high levels of insulin would emit exactly the same EGFP signal as a low-expressing one. In any case, the literature suggests that, if anything, BMP-7 would rather decrease insulin expression [see for instance Bruun *et al*, “Inhibition of beta cell growth and function by bone morphogenetic proteins”, *Diabetologia*, 2014)].

9. Other points. Introduction: “...are more likely to reflect the native biology of the organ” not completely true. They are the closest system to the in vivo condition so far, but till which degree they reflect native biology of the organ is not clear.

R: We have reworded this statement accordingly in the Introduction.

10. Fig1b-d. Could this analysis be extended to 10 days? This would help to monitor the system better over 10-day experimental course presented in later figures.

R: The oxygen content is reset at every medium change. In other words, the models show what slices would go through for the 24h that follow every medium change.

11. Page 7: “... in transwells have less-defined boundaries (Fig. 2j) and with less defined boundaries.” Repeated wording.

R: We thank the reviewer for the clarification. We have corrected this in the manuscript.

12. Some figures miss scale bar.

R: There are no error bars in 3f because this is one chamber’s perfusion. There are 3 slices in it but there is only one chamber. We could not use more chambers because this was a T1D sample, of which there is an exceedingly scarce supply that has to be shared among all the consortium members. We had only a very few slices and a minimum of 3 slices/perfusion chamber is needed. Therefore, these data are representative of one run. This has been clarified in the legend.

13. Fig2K. Are all these cells alive? Is there any analysis that can show the live vs dead cells in endocrine compartments?

R: While we did not conduct live/dead staining in this context, all endocrine cell types quantified in fig. 2k were positive for their respective hormone (insulin, glucagon and somatostatin) and DAPI, suggesting nuclear integrity. However, it cannot be discarded that some cells may be expressing apoptotic genes. This caveat has now been added to the Results section [*“Long-term viability (...)”* Results section].

14. Fig3g, I, what is 3G, please indicate it in the legend.

R: 3G is 3 mM of glucose (low glucose). This has now been added to the legend.

15. Fig3i, where is the reference or control sample at day 1?

R: These experiments were done in the context of an nPOD-sponsored consortium where our lab was charged with the conduct of long-term studies, leaving acute measurements to other partners. Slices from each donor were generated centrally and distributed to each lab for specific experiments. While we could use a few donors for acute perfusion experiments whenever slices were generated in Miami (fig. 3a), all acute Calcium imaging experiments were conducted in other labs. This is why acute is not shown in this graphic. At any rate, the entire second half of the figure basically aims at confirming that culture doesn't change significantly the expected physiology of glucose-mediated insulin secretion. This is now indicated in the manuscript [*“Enhanced oxygenation is supportive (...)”* Results section].

16. Fig3K-m, here only shows that the experiment technically works. I found the actual data regarding the PFC and TW cultures in the supplementary data. I would suggest to put those data in the main figure.

R: We have now upgraded the placement of suppl. fig. 3 data to the main figure and updated the legend accordingly [*“Enhanced oxygenation is supportive (...)”* Results section].

17. The experiment on mouse slices is done using pancreas from 6-8-week-old mice. This is relatively young age for mice. Therefore, the pancreatic lineage might still retain higher degree of plasticity. I am curious if old animals also present the formation of new beta cells when treated with BMP-7 in the slice culture.

R: We agree with the reviewer that conducting such experiments in mice of different ages would present us with a wealth of information on the age-related dynamics of β -cell regeneration. This is just one of the countless potential applications of our technique that we envision and hope will be explored not only by us, but by the rest of the scientific community once our findings are disseminated. However, we believe that such experiments fall beyond the scope of this initial communication.

18. Page13: “... due to the treatment of slices with BMP7.” They mean “alloxan”?

R: Yes, we thank the reviewer for noticing this error, which is now corrected.

19. Fig4c, is the lower picture cropped from the upper part? For me it doesn't seem to be.

R: The lower panel of Fig. 4c is a high-magnification confocal imaging of the same region shown in the upper panels by standard fluorescence microscopy. It is not a digital cropping, but a brand-new image taken immediately using a confocal microscope after the new green signal was detected by daily fluorescence examination. The region features a large pancreatic duct with a very characteristic shape that made it unmistakable. We have added the clarification “Higher magnification confocal imaging *of the same region*” in the text.

20. Fig4i. The green signal is barely detectable. The same for Fig4f.

R: The version of the manuscript available to the reviewer features a low-resolution pdf file. The original files, which will be provided for publication-grade images, show the green signal more clearly, but the panel size will remain a limitation. This is a higher resolution example showing the green channel (third picture) at day 6 in fig. 4f:

21. Page16: “We could also detect several regions of non- β -cells transitioning to EGFP-expressing cells, mirroring previous observations using mouse slices (Fig. 5a).” I don't see this transitioning in the picture. High magnification and resolution are required.

R: While we did detect transitions similar to those observed in mice (see example below, data not shown), the reviewer is correct that figure 5a does not offer enough resolution to substantiate the claim at the single-cell level. High-magnification confocal microscopy is not an option for the continuous monitoring of these phenomena: Since we do not know where any given green cell is going to appear, low-magnification panoramic imaging is required to capture the entirety of the slice. We are presently exploring non-confocal high-resolution alternatives for the real-time, single-cell tracking of regeneration. In the meantime, we have removed this sentence from the text and acknowledged this limitation of the analysis, which is still valid for the overall quantification of fluorescence changes at the tissue level.

22. Fig5d, is not mentioned within the text.

R: This is now corrected.

23. The conclusion section especially on neogenesis is too strong. It needs to be written more based on the actual data.

R: We have reworded the discussion according to the recommendation of the reviewer.

REVIEWER 3

NOTE: corrections in the manuscript are highlighted in yellow.

We thank reviewer 3 for highlighting the methodological novelty of our study, and for the suggestions to improve the accuracy of our claims. Below we address the specific comments/suggestions:

- 1. Figure 1: Authors state that parameters used to model oxygen consumption are estimated based on values reported in references 15. The data in reference 15 however reports data from isolated/cultured porcine acinar cells and islets kept in culture from 0 to 8 days. It's not clear why this data set was chosen to approximate/model oxygen consumption in intact human slices given clear differences in tissue handling and also reported differences between porcine and human islets/exocrine tissue. In addition, lactate production (panel e) data is presented (in hours) which is inconsistent with data presentation in other graphs (i.e. in days). Furthermore, authors need to include units for Fig 1e and 1f (i.e. lactate production and glucose consumption). Finally, the claim that ‘enhanced oxygenation’ preserves oxidative phosphorylation is not supported by presented data. Decreased “glucose consumption” in PFC group can be attributed to numerous factors and cellular changes and for example may reflect attenuated metabolic rate or changes in cellular proliferation/survival. Authors need to perform additional experiments to directly demonstrate enhanced mitochondrial function/oxygen consumption/oxidative metabolism or use glucose carbon isotope techniques to demonstrate enhanced glucose oxidation under PFC condition.***

R: We appreciate the reviewer's comments regarding our use of porcine acinar OCR as a gauge of oxygen consumption rate used in our finite element modeling. We agree that it is difficult to determine an exact rate given the differences in tissue handling and culture time, but feel that our choice of consumption rate is justified. Like the porcine tissue in the cited paper, the slices are cultured long term. In the cited paper, the islet consumption rate ranged from $1.5 \times 10^{-2} \text{ mol/m}^3 \text{ s}^{-1}$ to $2.6 \times 10^{-2} \text{ mol/m}^3 \text{ s}^{-1}$. The acinar tissue showed similar variability ranging $1.5 \times 10^{-2} \text{ mol/m}^3 \text{ s}^{-1}$ to $2.2 \times 10^{-2} \text{ mol/m}^3 \text{ s}^{-1}$. The observed differences between islet and acinar tissue in this article were surprisingly marginal, with a maximal 44% at D0 and minimal 13% at D1. All of these values were also well within the range of reported human islet oxygen consumption rates, which show great range in publication from values in the high $10^{-3} \text{ mol/m}^3 \text{ s}^{-1}$ to maximal $6-7 \times 10^{-2} \text{ mol/m}^3 \text{ s}^{-1}$. Our preferred rates of reference are those published by Klearchos Papas *et al* (author on the porcine reference of the manuscript), such as (1) <https://doi.org/10.1111/xen.12432>, PMID: 30052287 examining the OCR of porcine pancreatic islets from pigs of different ages, and his seminal work in (2) <https://doi.org/10.1002/bit.21486>, PMID: 17497731, on islet consumption rate. In all cases, the rates we selected for our finite element modeling were an average of the porcine acinar tissue rates over the culture period.

Finally, we modeled the oxygen distribution using a parametric sweep of tissue oxygen consumption rates varying from $0.9 \times 10^{-2} \text{ mol/m}^3 \text{ s}^{-1}$ - $5 \times 10^{-2} \text{ mol/m}^3 \text{ s}^{-1}$, looking at the effect of each culture system on the tissue anoxia/hypoxia at each consumption rate. Ultimately, the figure displayed the modelling outcomes that best matched the differences in viability/function observed in the functional/viability assays, the goal of predictive finite element modeling. It was not

unexpected to find that this occurred within the range of oxygen consumption rates described by others and us.

To further address the comments, we tried unsuccessfully to measure the oxygen consumption rate of the pancreatic slices (in the lone donor sample that we were able to secure after receiving these critiques) in the stirred chamber systems described by Papas and ourselves. We found that the slices did not perform well in the chambers, floating to the top of the sealed region and not forming the homogenous distribution of tissue seen with cells. We plan to keep trying to measure OCR in cultured slices from human pancreata to improve our future models.

The presentation of the lactate production data in hours was a mistake, and we thank the reviewer for noticing it. It has now been corrected to days, similar to that of glucose consumption rate.

The units for figs. 1e and f are μg of lactate/glucose/ μg of DNA. We thank the reviewer for detecting this omission, which is now corrected.

We also agree with the reviewer that, as stated, our data on glucose consumption and lactate production would be merely suggestive that PFC-cultured slices favor oxidative phosphorylation (OP) over glycolysis. However, we can now produce additional lines of evidence to support our claim. We have now gathered ATP production data for $n=3$ slices cultured in either TW or PFC for 10 days. The average ATP production in TW slices was $21.9 \text{ pm}/\mu\text{g}$ of protein, compared to $57.7 \text{ pm}/\mu\text{g}$ in those cultured in PFC, which is statistically insignificant ($P > 0.05$). However, the production of ATP (pm)/ μg of glucose, normalized by μg of protein, was 38.5 in the PFC group vs. 6.8 in the TW group (a ~6-fold increase). In short, both groups have comparable production of ATP, but the slices in PFC make it with 6 times less amounts of sugar. Therefore, it follows that the latter are differentially favoring the most efficient glucose utilization pathway (OP). We also show now significantly higher uptake of a fluorescent derivative of glucose (2-NBDG) by slices cultured in TW vs. those in PFC:

Quantification of the mean fluorescence intensity (MFI)/DAPI⁺ cells in both groups (0.53 ± 0.15 for the transwells vs. 0.075 ± 0.06 in PFC dishes, a 7-fold increase) confirms that the

differences between the two groups are statistically significant ($P=0.02$). The lower glucose uptake in the PFC group was not due to lower viability of the tissue—in fact, the opposite is true, as there was a 2.6-fold increase in the ratio of propidium iodide (PI, cell death marker): DAPI in the transwell (0.40 ± 0.04) vs. the PFC (0.14 ± 0.06) groups ($P=0.007$). This observation further supports the hypothesis that the slices cultured in transwells require a higher glucose uptake to keep up with the tissue energy demands.

Taken together, we show that TW-cultured slices have a higher glucose consumption rate (GCR), confirmed by immunofluorescence detection of glucose uptake, than those cultured in PFC dishes. Since both groups produce the same amount of ATP, but the TW slices require 6 times more glucose to achieve the same, the evidence strongly suggests that the control group has a higher glycolysis: OP ratio. However, we agree that more comprehensive studies including carbon isotope techniques to demonstrate enhanced glucose oxidation under PFC would be necessary to unequivocally demonstrate our hypothesis.

- 2. Figure 2. Given focus of the manuscript on endocrine pancreas biology, authors should perform additional studies of endocrine pancreas viability/expression in PFC vs. TW culture conditions. This should include examination of cell turnover (proliferation and apoptosis) in beta and alpha cells. This will be particularly important given the focus on beta cell regeneration and turnover. In addition, author should examine expression of key maturation markers for beta (e.g. MAFA, NKX6.1, PDX-1 etc) and alpha (e.g. ARX) cells. This is important given dramatic increases in insulin secretion reported in Fig. 3 under PFC conditions. Finally, since authors suggest enhanced oxygen delivery and improved mitochondrial oxidation in PFC group, staining for hypoxia-regulated transcription factors (e.g. HIF1, c-JUN etc) and/or markers of mitochondrial integrity will provide supporting evidence for the stated hypothesis.***

R: In general, we appreciate the suggestion of additional studies that would undoubtedly enhance the depth of our manuscript. In this context, it is important to stress that this work was conducted within a consortium, the nPOD Slice Working Group, which originally involved 5 teams. Our manuscript lists the leaders of these (Speier, Atkinson, Pugliese, Caicedo), who have had an active input throughout. Organs for slicing were centrally processed either in our lab (Miami, FL) or the nPOD lab (Gainesville, FL). Over the past 3 years, only a limited number of slices/donor organ was made available to each one of these teams for specific experiments. Even the slices that we generated in our lab were distributed to the other consortium teams as well. Thus, for each donor, we had to carefully maximize the number of assays to be performed; and, when slices were insufficient, prioritize such assays. In other words, the scope of our research has been dictated to a large extent by the available number of slices and—needless to say—the irregularity of offers of human pancreata for research. To this, we have to add that not all samples were suitable for slicing, particularly if the pancreas was fatty (a very common occurrence) or fibrotic.

This is all to say that, while the reviewer proposes some very worthy experiments to expand even further the breadth of our findings (many of which already in our pipeline), conducting all of them with an appropriately powered number of samples would take a very long and unpredictable amount of time. Our manuscript is intended as a first report of a new technology to study the human pancreas, not as a comprehensive opus that provides all the answers. We hope that, by making this

initial report available to the scientific community, others will also be able to exploit our technology to the fullest extent.

Having said that, we have been able to secure one additional human pancreatic sample suitable for slicing, and we have used the limited resulting slices to run some of the experiments suggested by the reviewers. Owing to the fact that these results are not appropriately powered, we present these findings to reviewers and refer to them in the manuscript as part of supplemental information. One of such experiments is the calculation of HIF-1 α in both groups after a 24h culture period in both TW- and PFC-cultured human slices, as suggested by the reviewer. We did this by ELISA, normalizing the results to total tissue protein content. The results of $n=3$ slices/group are shown below [$P<0.001$ between biological replicates (slices) of the same sample]. This is now in **fig. S1a**.

HIF-1 α is elevated at the protein level in the slices cultured in TW vs. those cultured in PFC, as expected from our previous determination that the former exhibit a significant degree of anoxia.

As indicated before, while we believe that endocrine cell turnover experiments are truly worth conducting, doing so with an appropriately powered number of samples at this stage would take many months without any guarantee of a steady supply of human pancreata of sufficient quality. We have now modified the Discussion to reflect the importance of conducting such experiments in the future.

To partially address the comment of the reviewer, we have performed a qRT-PCR analysis of a panel of key pancreatic markers on one additional human pancreatic sample that we have been able to analyze. This is now shown in **Fig. S1d**.

- 3. Figure 3. Authors report enhanced glucose-stimulated insulin secretion under perfusion conditions with 10-day culture using PFC dishes. Although this data is impressive, there are number of questions that arise from these studies. Firstly, contrary to what authors state, none of the groups display “classic biphasic pattern” of glucose-stimulated insulin release. In PFC group, insulin concentrations continuously rise during 16mM glucose period without clear 1st/2nd phase**

distinctions. The physiological significance of pancreas slice perfusion will be reinforced by data demonstrating physiological release of other important endocrine hormones such as glucagon and somatostatin. Finally, PFC group has a significant increase in glucose-independent insulin release to KCl. KCl is commonly used as a measure of maximal insulin output and used as a marker of total beta cell mass/numbers. Authors should comment on the reason for increased KCl-stimulated insulin release in PFC group despite comparable levels of insulin positive cells reported in sections from PFC vs. TW slices.

R: We have reworded the manuscript to reflect the abnormality of the 16 mM stimulation phase. While the suggestion of calculating glucagon and somatostatin release is very interesting, we decided to dedicate the limited number of slices available mostly to beta cells studies—which, after all, are the ones whose regeneration we aim at studying. We are already collaborating with another nPOD team whose main interest is alpha cell biology, and we look forward to elucidating these and other related questions in the future.

Regarding the comment on KCl stimulation, we agree with the reviewer that our observation merits additional discussion. Since our results are normalized by the overall number of DAPI⁺/INS⁺ cells/slice, it follows indeed that β -cells cultured long-term in PFC exhibit a higher per-cell secretion of insulin. We speculate that this is due to the suboptimal wellbeing of islets in control conditions. Daily exposure to hypoxic conditions may result in a derivation of the β -cell resources to survival rather than insulin production. It is well documented that β -cells decrease insulin production in response to physiological stress (see, for example, Swisa et al, 2017; Front Genet. 2017 Feb 21;8:21. doi: 10.3389/fgene.2017.00021. *Metabolic Stress and Compromised Identity of Pancreatic Beta Cells*), and we add now discussion to this effect.

- 4. Figures 4 and 5. Data on longitudinal tracking of purported beta cell regeneration in mouse/human slices is intriguing, but is largely preliminary and not supportive of main conclusions of the study. Firstly, to prove generation of “newly” formed beta cells (with BMP-7) authors need to show clear increase in overall beta cell numbers (or other measures of insulin + area). These data need to show that newly formed cells express key beta cell maturation transcription factors (e.g. NKX6.1, PDX1 etc) as well as other markers of functionality. Demonstrating that treatment with BMP-7 also positively alters insulin secretion in perfusion setting (especially KCl-mediated insulin release) will also significantly strengthen the argument that beta cell neogenesis has occurred in this model. In addition, the concept of beta cell regeneration implies replacement of lost beta cells, however current study in human slices examines response of otherwise “healthy” beta cells to a supra-physiological stimuli of BMP-7. Authors need to clearly acknowledge that there are limitations of this approach in human tissue and the study of human beta cell physiological neogenesis is not possible using this model. Indeed, repeated studies in mice and humans have demonstrated lack of any evidence for physiological occurrence of beta cell neogenesis.***

R: We agree that the data are preliminary. As also indicated to the other reviewers, multiple other lineage-tracing strategies would be needed to confirm endocrine cell neogenesis. However,

we do show a clear, statistically significant increase in the area of cells labeled green upon activation of the insulin promoter in figures 4 and 5.

Again, the scarcity in the number of slices available for any given experiment has often limited our ability to conduct extensive IF characterization. We show now by qRT-PCR that all key functional pancreatic endocrine and exocrine markers that we analyzed are upregulated in the PFC vs. the TW group (**Fig. S1d**), but future studies will be necessary to establish whether newly formed insulin-producing cells are functional. This caveat is indicated in the discussion.

The reviewer indicates that “*Demonstrating that treatment with BMP-7 also positively alters insulin secretion in perfusion setting (especially KCl-mediated insulin release) will also significantly strengthen the argument that beta cell neogenesis has occurred in this model*”. We agree with this assessment, but since these pancreatic slices already contain native islets, and the newly created insulin-expressing cells represent proportionally only a very small fraction, the interpretation of KCl-mediated insulin release assays would be very challenging. We aim at conducting future calcium imaging experiments to indirectly determine glucose responsiveness in newly created insulin-expressing cells, but this will require a major overhaul of the existing lineage tracing setting. We hope these will be the subject of a future report, done in collaboration with another nPOD group, focused on the physiology of the new INS⁺ cells.

The reviewer also states that “*the concept of beta cell regeneration implies replacement of lost beta cells. However, current study in human slices examines response of otherwise ‘healthy’ beta cells to a supra-physiological stimuli of BMP-7*”. We agree that regeneration would normally occur in response to β -cell loss, and typically from self-replication of preexisting mature cells. However, it has been extensively documented that formation of new β -cells may also occur in response to stress from different types of pancreatic cells. In this case, we conduct mechanical slicing of an organ that has been ischemic for a considerable amount of time, as is invariably the case for most donor organs. It stands to reason that these tissues are exposed to stress, which has been associated to neogenesis from ductal regions regardless of the loss of β -cells. For instance, using the duct ligation model, Bouwens and colleagues showed a decade ago that adult ductal cells proliferated in a progenitor-like fashion and gave rise to all endocrine cell types, both *in vivo* and *in vitro* (Xu et al., 2008) (PMID: 18243096). This report was so compelling that even the original proponents of the “self-replication only” hypothesis published a commentary later that year acknowledging the presence of “facultative endocrine progenitor cells” in the exocrine pancreas (Dor and Melton, 2008) (PMID: 18243094). The use of proinflammatory cytokines has also been shown to activate progenitor-like cells in ducts (Valdez et al., 2016) (PMID: 27068459). Similarly, subtotal pancreatectomy and gastrin treatment have been reported to induce ductal dedifferentiation and β -cell neogenesis (Tellez and Montanya, 2014) (PMID: 25122000). Even more recently, the Kulkarni lab has shown that human ductal cells contribute to β -cell compensation in insulin resistance (Dirice et al, *JCI Insight*, 2019) (PMID: 30996131). There is a common misperception that the debate has been settled against adult β -cell neogenesis, but current views are much more nuanced, with countless other more recent reports showing a developmental continuum in the ductal tree and facultative activation of progenitor-like cells under significant stress. We recently reviewed the status of this debate in Qadir et al, 2018 (*Trends in Endocrinology and Metabolism*, PMID: 30502039). The present manuscript, however, is merely intended as the presentation of a tool to study these phenomena in a unique human culture setting. We acknowledge that no definitive conclusions on regeneration can be drawn from these data, but they present a powerful technique that is likely to shed much-needed light on this area in future studies.

Reviewers' Comments:

Reviewer #1:

Remarks to the Author:

To reiterate, I do think that this paper is a major breakthrough in pancreatic tissue long term culture of up to a remarkable 10 days, which is of sufficient duration for numerous applications in pancreatic islet and exocrine biology, and here the islet capability was applied to understanding neogenesis of beta cells using viral mediated lineage tracing in normal pancreas. I strongly believe that this technical capability is extremely important and useful to the large community of pancreatic investigators and should be disseminated.

In the revised manuscript and rebuttal, these workers have satisfactorily addressed almost all of my concerns, although the new exocrine data is not perfect with regards to histological details, this is not the major point of this work.

There must be an error in Fig. 3t using 10 MICROMolar CCK8, which is highly toxic, and likely they were using 10 (or 100) PICOmolar CCK8.

Reviewer #2:

Remarks to the Author:

The authors have addressed all of my concerns either experimentally or by adding/modifying texts. Therefore, I believe that the current version of the manuscript is suitable for publication.

Reviewer #3:

Remarks to the Author:

This reviewer appreciates and acknowledges the hard work of authors to provide additional data related to modeling of oxygenation, staining for beta cell maturation markers, glucagon secretion data etc. It is noteworthy that the authors have made a genuine attempt to address these questions.

RESPONSE TO REVIEWERS

Reviewer 1

To reiterate, I do think that this paper is a major breakthrough in pancreatic tissue long term culture of up to a remarkable 10 days, which is of sufficient duration for numerous applications in pancreatic islet and exocrine biology, and here the islet capability was applied to understanding neogenesis of beta cells using viral mediated lineage tracing in normal pancreas. I strongly believe that this technical capability is extremely important and useful to the large community of pancreatic investigators and should be disseminated.

In the revised manuscript and rebuttal, these workers have satisfactorily addressed almost all of my concerns, although the new exocrine data is not perfect with regards to histological details, this is not the major point of this work.

There must be an error in Fig. 3t using 10 MICROMolar CCK8, which is highly toxic, and likely they were using 10 (or 100) PICOMolar CCK8.

R: We thank reviewer 1 for his/her appreciation of the impact of our work, and for noticing this mistake, which has been corrected in the final version of the manuscript. His/her insightful comments have greatly improved our manuscript.

Reviewer 2

The authors have addressed all of my concerns either experimentally or by adding/modifying texts. Therefore, I believe that the current version of the manuscript is suitable for publication.

R: We thank reviewer 2 for his/her input through the process, and for his/her invaluable comments and suggestions, owing to which our manuscript is stronger.

Reviewer 3

This reviewer appreciates and acknowledges the hard work of authors to provide additional data related to modeling of oxygenation, staining for beta cell maturation markers, glucagon secretion data etc. It is noteworthy that the authors have made a genuine attempt to address these questions.

R: We thank reviewer 3 for his/her input through the process, and for his/her invaluable comments and suggestions, owing to which our manuscript is stronger.